# Fibroblast-induced mammary epithelial branching depends on fibroblast contractility

**Jakub Sumbal**[1,2,3], **Silvia Fre**[2], **Zuzana Sumbalova Koledova**[1¤]*

**1** Masaryk University, Faculty of Medicine, Department of Histology and Embryology, Brno, Czech Republic,
**2** Institut Curie, Laboratory of Genetics and Developmental Biology, INSERM U934, CNRS UMR3215, PSL Université Paris, Paris, France, **3** Sorbonne Université, Collège Doctoral, Paris, France

¤ Current address: Institute of Molecular Genetics of the Czech Academy of Sciences, Prague, Czech Republic

* koledova@med.muni.cz, zuzana.sumbalova-koledova@img.cas.cz

**Data Availability Statement:** All relevant data are within the paper and its Supporting Information files.

**Funding:** This work was supported by grants from the Grant Agency of Masaryk University (MU)

## Abstract

Epithelial branching morphogenesis is an essential process in living organisms, through which organ-specific epithelial shapes are created. Interactions between epithelial cells and their stromal microenvironment instruct branching morphogenesis but remain incompletely understood. Here, we employed fibroblast-organoid or fibroblast-spheroid co-culture systems and time-lapse imaging to reveal that physical contact between fibroblasts and epithelial cells and fibroblast contractility are required to induce mammary epithelial branching. Pharmacological inhibition of ROCK or non-muscle myosin II, or fibroblast-specific knockout of *Myh9* abrogate fibroblast-induced epithelial branching. The process of fibroblast-induced branching requires epithelial proliferation and is associated with distinctive epithelial patterning of yes associated protein (YAP) activity along organoid branches, which is dependent on fibroblast contractility. Moreover, we provide evidence for the in vivo existence of contractile fibroblasts specifically surrounding terminal end buds (TEBs) of pubertal murine mammary glands, advocating for an important role of fibroblast contractility in branching *in vivo*. Together, we identify fibroblast contractility as a novel stromal factor driving mammary epithelial morphogenesis. Our study contributes to comprehensive understanding of overlapping but divergent employment of mechanically active fibroblasts in developmental versus tumorigenic programs.

## Introduction

Efficient formation of large epithelial surfaces in limited organ volumes is achieved through branching morphogenesis [1]. The underlying processes of epithelial morphogenesis, including epithelial cell proliferation, migration, intercalation, differentiation, and death, are regulated by both internal genetic programs as well as external cues provided by systemic signals (such as hormones) and local organ-specific microenvironment [1–3]. The mammary gland is the ideal tissue paradigm for stochastically branching epithelia. Mammary morphogenesis starts in the embryo, but the majority of branch bifurcations and ductal elongation takes place postnatally during puberty. During this time epithelial morphogenesis is driven by terminal

(https://gamu.muni.cz/; grants no. MUNI/G/1446/
2018, MUNI/G/1775/2020 to Z.S.K., MUNI/A/1398/
2021 and MUNI/A/1301/2022), from Internal Grant
Agency of Faculty of Medicine MU (MUNI/11/SUP/
06/2022 to Z.S.K. and MUNI/IGA/1314/2021 to J.
S.), from Foundation pour la Recherche Médicale
(FRM) (https://www.frm.org/; grant no. "FRM
Equipes" EQU201903007821 to S.F., the
Association for Research against Cancer (ARC)
(https://www.fondation-arc.org/the-fondation-arc/;
grant no. ARCPGA2021120004232_4874) to S.F.,
and from Czech Science Foundation (GAČR)
(https://gacr.cz/; grant no. GA23-04974S to Z.S.K.).
J.S. is supported by Barrande Fellowship (Ministry
of Education, Youth and Sports; https://www.
msmt.cz/), Fondation pour la Recherche Médicale
(https://www.frm.org/; grant no.
FDM202106013570), and by Brno PhD. Talent
Scholarship, funded by the Brno City Municipality
(https://www.jcmm.cz/projekt/brno_phd_talent/).
The funders had no role in study design, data
collection and analysis, decision to publish, or
preparation of the manuscript.

**Competing interests:** The authors have declared
that no competing interests exist.

**Abbreviations:** 3SB, 3D staining buffer; CAF,
cancer-associated fibroblast; ECM, extracellular
matrix; FBS, fetal bovine serum; FGF2, fibroblast
growth factor 2; FGFR, FGF receptor; IHC-IF,
immunohistochemistry-immunofluorescence; ROI,
region of interest; TEB, terminal end bud; YAP, yes
associated protein.

end buds (TEBs), bulb-shaped structures containing proliferative stratified epithelium that invades the surrounding mammary stroma [4].

The microenvironment of the mammary epithelium is a dynamic entity that consists of extracellular matrix (ECM) and stromal cells, including fibroblasts. Fibroblasts lay adjacent to the epithelium and have been well recognized as master regulators of mammary epithelial morphogenesis during puberty through production of growth factors [5–9] and ECM molecules [5,7,9–14] necessary for mammary epithelial growth and branching [15]. However, the dynamics of the epithelial–fibroblast interactions during mammary branching morphogenesis as well as whether fibroblasts contribute to shaping of mammary epithelium through additional mechanisms have remained unknown.

Microenvironment of several developing organs has been shown to govern epithelial patterning by dynamic cues of mechanically active cells. Dermal cells in chick skin determine feather buds by mechanical contraction [16], intestinal vilification is dependent on compression by smooth muscle cells [17], and embryonic lung mesenchyme promotes epithelial bifurcation by mechanical forces [18–20]. However, it has not been elucidated whether the mammary microenvironment contains an instructive component of mechanically active cells as well.

To answer this question, we performed live imaging and functional analysis of co-cultures of primary mammary epithelial organoids (isolated epithelial fragments with in vivo like architecture consisting of inner luminal and outer myoepithelial (basal) cells) with primary mammary fibroblasts. Analogously to primary mammary organoids treated with fibroblast growth factor 2 (FGF2), a well-established model of mammary branching morphogenesis driven by paracrine signals [21], our in vitro co-culture model provides a unique window into fibroblast–epithelial interactions during pubertal mammary branching morphogenesis. It enables visualization of stromal fibroblasts during dynamic morphogenetic processes, which are otherwise largely inaccessible in vivo due to light-scattering properties of mammary adipose tissue. In this work, we show that physical contact between fibroblasts and epithelial cells, and actomyosin-dependent contractility of fibroblasts are required for branching morphogenesis. We demonstrate successful reconstitution of budding morphogenesis by 3D co-culture of contractile fibroblasts in breast cancer spheroids that normally do not form buds. Moreover, by combining contractile fibroblasts with strong proliferative signals we reproduce TEB-like branching morphogenesis in organoid cultures and reveal localization of contractile fibroblasts around TEBs in the mammary glands, suggesting a role for fibroblast contractility in vivo. Our results reveal a novel role of fibroblast contractility in driving epithelial branching morphogenesis.

## Results

### Fibroblast-induced branching of organoids does not reproduce FGF2-induced budding

To uncover the role of fibroblasts in epithelial morphogenesis, we investigated differences between organoid budding induced solely by paracrine factors (using primary mammary organoids exposed to exogenous FGF2) and organoid branching induced by fibroblasts (using organoids co-cultured with primary mammary fibroblasts in the absence of any exogenous growth factor). Addition of FGF2 or fibroblasts to mammary organoid cultures both induced branching of epithelial organoids (Fig 1A and 1B and S1 Movie) but examination of the resulting organoid morphogenesis revealed important differences in dynamics and epithelial architecture in the 2 conditions. First, organoids co-cultured with fibroblasts developed bigger branches, but the branches were less numerous (Fig 1A, 1C and 1D). Second, they branched

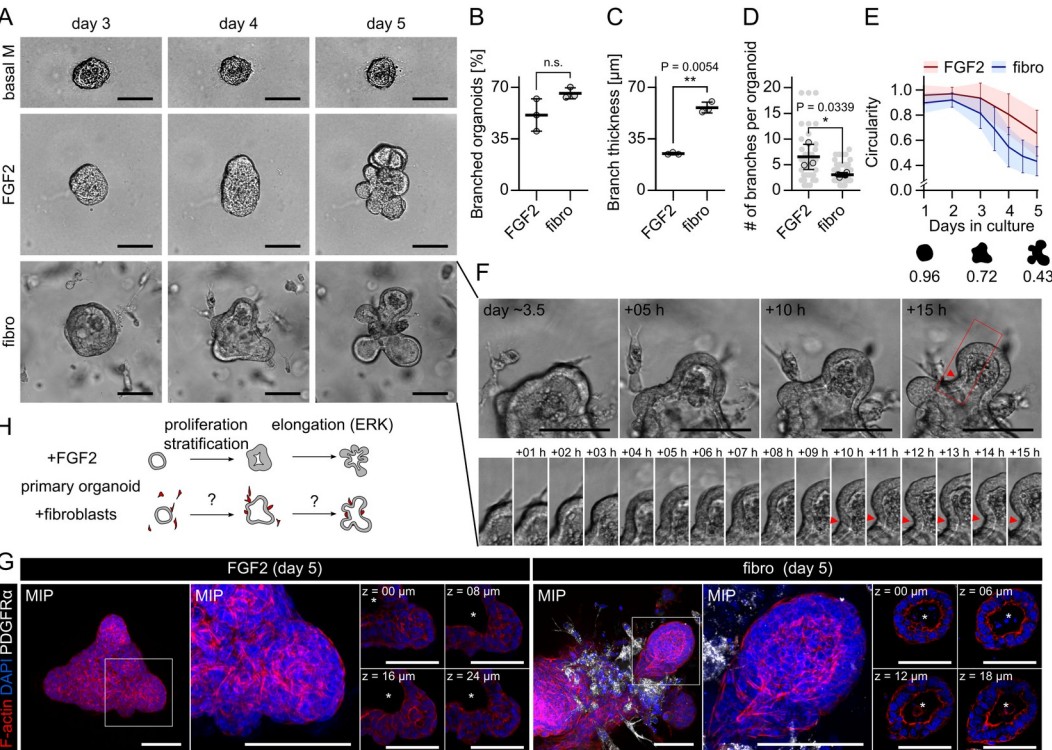

**Fig 1. Fibroblast-induced branching of organoids does not reproduce FGF2-induced budding.** (A) Snapshots from time-lapse imaging of primary mammary organoids in basal organoid medium (basal M) without any FGF supplementation (top), basal M with FGF2 (middle), or co-cultured with primary mammary fibroblasts (fibro) in basal M with no FGF supplementation (bottom). Scale bar: 100 μm. Full videos are presented in S1 Movie. (B) Quantification of percentage of branched organoids per all organoids in the conditions from (A). The plot shows mean ± SD, each dot represents biologically independent experiment, *n* = 3. Statistical analysis: Two-tailored *t* test. (C) Quantification of branch thickness from experiments in (A). The plot shows mean ± SD, each dot represents a biologically independent experiment, *n* = 3. Statistical analysis: Two-tailored *t* test. (D) Quantification of number of branches per branched organoids in conditions from (A). The plot shows mean ± SD, each lined dot shows mean from each experiment, each faint dot shows single organoid measurement, *n* = 3 biologically independent experiments, *N* = 20 organoids per experiment. Statistical analysis: Two-tailored *t* test. (E) Quantification of organoid circularity in conditions from (A). The lines represent mean, the shadows and error bars represent ± SD, *n* = 3 biologically independent experiments, *N* = 20 organoids per experiment. The schemes show representative shape of indicated circularity. (F) Detailed images of branch development in co-culture with fibroblasts from (A). Scale bar: 20 μm. (G) MIP of F-actin (red), DAPI (blue), and PDGFRα (white) in organoid with exogenous FGF2 or with fibroblasts (fibro). Zoom-in area from the box is depicted as MIP and single z slices. The asterisks denote lumen. Scale bar: 100 μm. (H) A scheme depicting differences between organoid budding induced by exogenous FGF2 and organoid branching in a co-culture with fibroblasts. The data underlying the graphs shown in the figure can be found in S1 Data. FGF2, fibroblast growth factor 2; MIP, maximum intensity projection.

half-day to 1 day earlier than organoids treated with FGF2 (Fig 1A and 1E) and the branches were developed rapidly, including the development of negative curvature at the root of the branch (Fig 1F). Third, while FGF2-induced epithelial branching involved epithelial stratification as previously reported [21], co-culture with fibroblasts did not perturb the epithelial bilayer with its lumen (Fig 1A and 1G). These results suggest that fibroblasts and exogenous FGF2 drive organoid branching by different mechanisms.

While the mechanism of FGF2-induced organoid budding was previously described in detail to begin with epithelial proliferation and stratification [22] followed by budding and ERK-dependent and proliferation-independent bud elongation [23], how fibroblasts induce organoid branching remains unanswered (Fig 1H).

## Endogenous paracrine signals are not sufficient to induce organoid branching in co-cultures

The ability of exogenous FGF2 to promote organoid budding [21] (Fig 1A) well demonstrates the importance of paracrine signals for epithelial branching, although FGF2 amount used in in vitro branching assays likely exceeds physiological values in vivo. Therefore, we sought to determine, whether endogenous FGFs or other paracrine signals produced by fibroblasts in co-cultures [5,7] are sufficient to drive organoid branching. First to test the involvement of FGF signaling, we inhibited either FGF receptors (FGFRs) using SU5402, or ERK, a common signaling node of all receptor tyrosine kinases using U0126. As expected, both inhibitors abolished branching induced by exogenous FGF2 (Fig 2A and 2B). However, in the co-cultures with fibroblasts, the same concentration of inhibitors did not abolish branching, albeit slightly reduced organoid growth (Fig 2A and 2B), suggesting that paracrine signaling via FGFR-ERK pathway is not the only mechanism driving organoid branching.

To probe the involvement of other paracrine signaling pathways in fibroblast-induced organoid branching, namely to test if fibroblast paracrine signaling alone is sufficient to induce organoid branching, or if other mechanisms involving fibroblast–epithelial proximity or contact are involved, we performed an array of different types of organoid (co-)culture set-ups (Fig 2C, top). When we provided unidirectional fibroblasts-to-epithelium paracrine signals by culture of organoids in fibroblast-conditioned medium, no organoid branching was observed (Fig 2C and 2D). When we allowed bidirectional paracrine signals by co-culture of organoids with fibroblasts in the same well but the organoids and fibroblasts were separated by a transwell membrane or by a thick layer of Matrigel, no organoid branching was observed either (Fig 2C and 2D). However, when we allowed both paracrine signals and physical contact between organoids and fibroblasts by co-culturing them together either dispersed in Matrigel or as aggregates of fibroblasts on top of organoids embedded in Matrigel, we observed organoid branching (Fig 2C–2E). These results demonstrated an essential requirement of fibroblast–epithelium contact for fibroblast-induced organoid branching, thus revealing that fibroblast-secreted paracrine factors are not sufficient to initiate branching.

## MCF7-ras spheroids recapitulate fibroblast-induced branching of organoids

To further test the requirement of fibroblast–epithelium contact for fibroblast-induced epithelial branching, we developed a simpler co-culture system, where mammary fibroblasts were co-cultured with MCF7-ras breast cancer cell line spheroids (Fig 2F) instead of organoids from normal mammary epithelium. The advantage of MCF7-ras spheroids is that the spheroids grow constantly due to constitutively active RAS GTPase and unlike normal epithelium they do not respond to exogenous FGF2 (Fig 2G) or EGF (S1A Fig) by branching. We found that similarly to normal epithelium, MCF7-ras spheroids remained round in fibroblast co-cultures, which did not allow physical contact with fibroblasts, but developed numerous buds when physical contact with fibroblasts was allowed (Fig 2H and S1B Fig). These results demonstrated that fibroblasts are able to promote epithelial budding even in a system that is morphogenetically unresponsive to paracrine signals.

## Fibroblasts form physical contact with organoids

To gain more insights into the mechanism of fibroblast-induced organoid branching, we examined organoid branching in the dispersed co-cultures in more detail. A day-by-day analysis of the contacts between fibroblasts and organoids revealed that the contacts are established

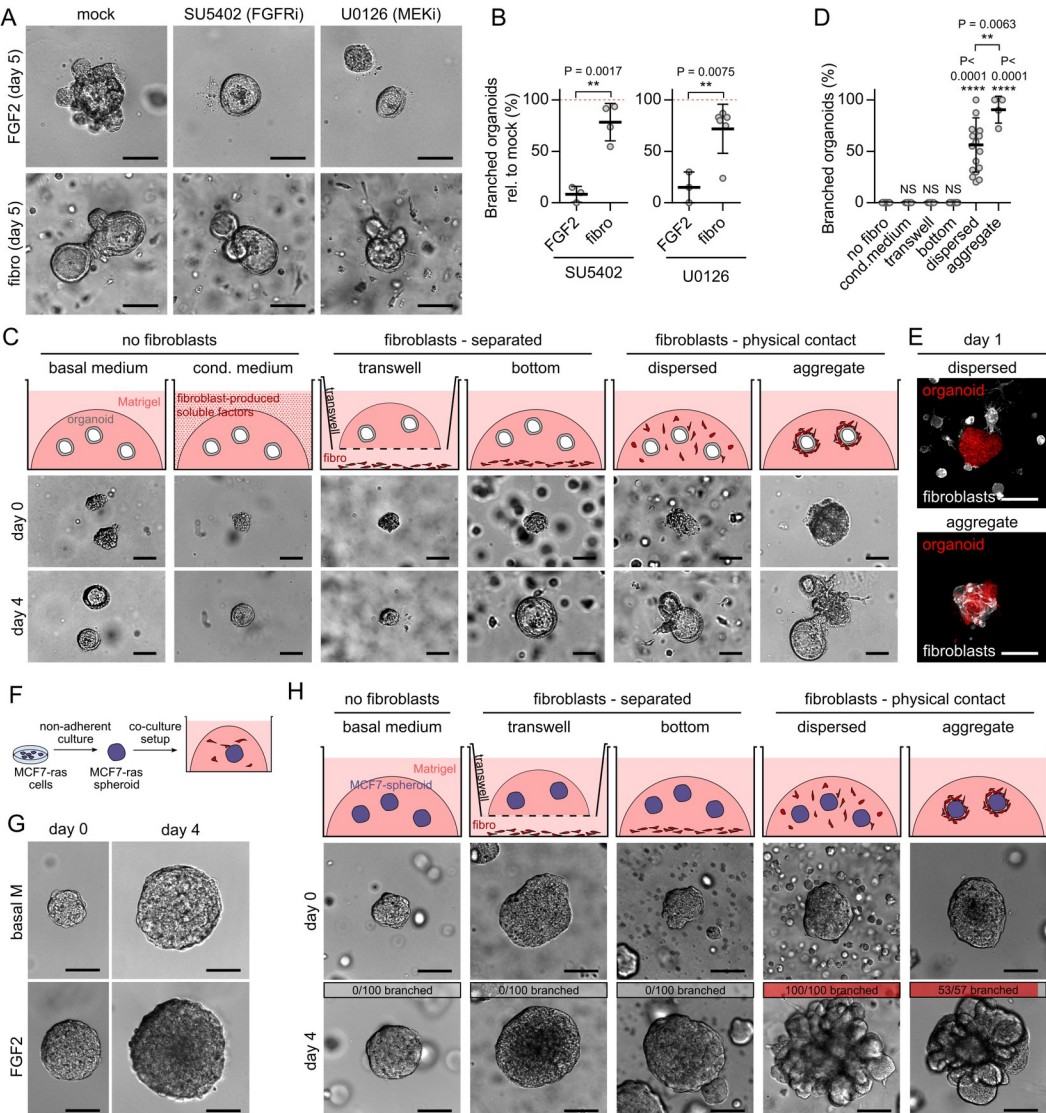

**Fig 2. Endogenous paracrine signals are not sufficient to induce organoid branching. (A)** Representative organoids cultured with exogenous FGF2 or with fibroblasts (fibro), treated with inhibitors of FGFR (SU5402) and MEK (U0126). Scale bar: 100 μm. **(B)** Quantification of branched organoid per all organoids, relative to mock. The plots show mean ± SD, each dot represents biologically independent experiment, *n* = 3–5, *N* = 20 organoids per experiment. Statistical analysis: Two-tailored *t* test. **(C)** Schemes and images on day 0 and day 4 of different organoid-fibroblast co-culture set-ups. Scale bar: 100 μm. **(D)** Quantification of organoid branching in different co-culture set-ups. The plot shows mean ± SD, each dot represents biologically independent experiment, *n* = 16 independent experiments for "no fibro," 5 for "cond. medium," 5 for "transwell," 12 for "bottom," 16 for "dispersed," and 4 for "aggregate," *N* = 20 organoids per experiment. Statistical analysis: Multiple *t* tests compared to control "no fibro," or indicated by the line. **(E)** Dispersed and aggregated co-culture of LifeAct-GFP fibroblast (white) and tdTomato organoid (red) at the beginning of the culture. Scale bar: 100 μm. **(F)** A scheme of MCF7-ras spheroid co-culture setup. **(G)** Representative MCF7-ras spheroids cultured in basal organoid medium (basal M) or basal M with exogenous FGF2. Scale bar: 100 μm. **(H)** Schemes and images on day 0 and day 4 of different spheroid-fibroblast co-culture set-ups. Top gray and red bars indicate proportion of branched spheroids out of all spheroids per condition, *n* = 3–5 independent experiments, *N* = 20 spheroids per experiment. Scale bar: 100 μm. The data underlying the graphs shown in the figure can be found in S1 Data. FGF2, fibroblast growth factor 2; FGFR, FGF receptor.

from the first day and thus precede the branching events that occur on days 3 and 4 (S2A–S2C Fig). To corroborate this finding, we performed a co-culture experiment with organoids labeled by tdTomato and GFP-tagged fibroblasts. The time-lapse movies confirmed that

fibroblasts came in close contact with the epithelium early during the co-culture and remained there during branching (Fig 3A and S2 Movie). Confocal imaging analysis revealed that fibroblasts (marked by PDGFRα) came in contact with all organoids (100% of 59 organoids analyzed in 3 independent biological replicates, S2D Fig) and contacted a larger proportion of the organoid middle sectional perimeter in round organoids than in branched organoids (S2E Fig).

On the branched organoids, fibroblasts were predominantly located around the necks of the nascent branches and sat directly in contact with the epithelium (Fig 3B and 3C). Immunofluorescence staining of epithelial markers revealed that fibroblasts formed contacts with KRT5 positive myoepithelial cells (Fig 3D and 3E). Transmission electron microscopy of the co-cultures confirmed the close proximity between the fibroblasts and the epithelium, with a thin layer of ECM in between (Fig 3F). Using immunostaining we detected laminin 5, a basal membrane component, between the organoid and the adjacent fibroblast (Fig 3G and 3H). These data suggest that fibroblasts form contacts with epithelium via ECM.

## Fibroblast-induced epithelial branching depends on fibroblast contractility

Based on observations from the time-lapse videos of organoids branched by fibroblasts (S1 and S2 Movies), we hypothesized that fibroblasts could constrict epithelium, folding it into branches. Immunofluorescence investigation of fibroblast-branched organoids revealed that fibroblasts in contact with the organoid formed a cellular loop, encircling the branch neck (Figs 4A and S3A–S3C and S3 Movie), and contained F-actin cables oriented mostly perpendicularly to the branch longitudinal axis (Fig 4A). Moreover, the fibroblasts stained positively for phosphorylated myosin light chain 2 (P-MLC2), a marker of active non-muscle myosin II (Fig 4B). Therefore, we examined the involvement of fibroblast contractility in fibroblast-induced organoid branching using small molecule inhibitors of non-muscle myosin II (blebbistatin) or ROCK1/2 (Y27632), 2 major nodes of cell contractility. The contractility inhibitors abrogated branching in co-cultures but did not inhibit organoid budding induced by exogenous FGF2 (Fig 4C and 4D; ROCK inhibition in FGF2-induced organoids even led to hyperbranched phenotype as previously described [21]). Similarly to organoid co-cultures, in the MCF7-ras spheroid co-culture model, spheroid budding was inhibited by addition of contractility inhibitors (S4A–S4D Fig). Importantly, the contractility inhibitors did not diminish the capability of fibroblasts to migrate towards and contact the organoid (S5A–S5D Fig), in agreement with previous reports that showed that fibroblast migration in 3D is not abrogated by non-muscle myosin II inhibition [7]. Noteworthy, addition of the contractility inhibitors on day 3 of the co-culture, when branches were already formed, led to retraction of formed branches (S6A–S6D Fig), suggesting a role of contractility in branch maintenance as well as initiation.

Because exogeneous treatment with pharmacological inhibitors in the culture medium affects both epithelial cells and fibroblasts, we genetically targeted exclusively in fibroblasts the contractility machinery gene myosin heavy chain 9 (*Myh9*), one of the 2 non-muscle myosin II heavy chain genes expressed in mammary fibroblasts (S7A and S7B Fig). Both siRNA-mediated *Myh9* knock-down in wild-type fibroblasts and adenoviral Cre-mediated knock-out in *Myh9*$^{fl/fl}$ fibroblasts led to a decrease of organoid branching in co-cultures (Figs 4E, 4F and S8A–S8E and S4 and S5 Movies). To analyze if *Myh9* knock-out affected the ability of fibroblasts to migrate towards and contact the organoid, we took advantage of the mosaic nature of the adenovirus-mediated gene delivery. We quantified the amount of GFP+ fibroblasts (transduced by either Ad-Cre-GFP or Ad-GFP) and GFP-fibroblasts (not transduced by either of the adenoviruses) and compared their migration towards and contact with epithelium. We found no differences in their migration and epithelium-contacting abilities (S9A and S9B Fig). These

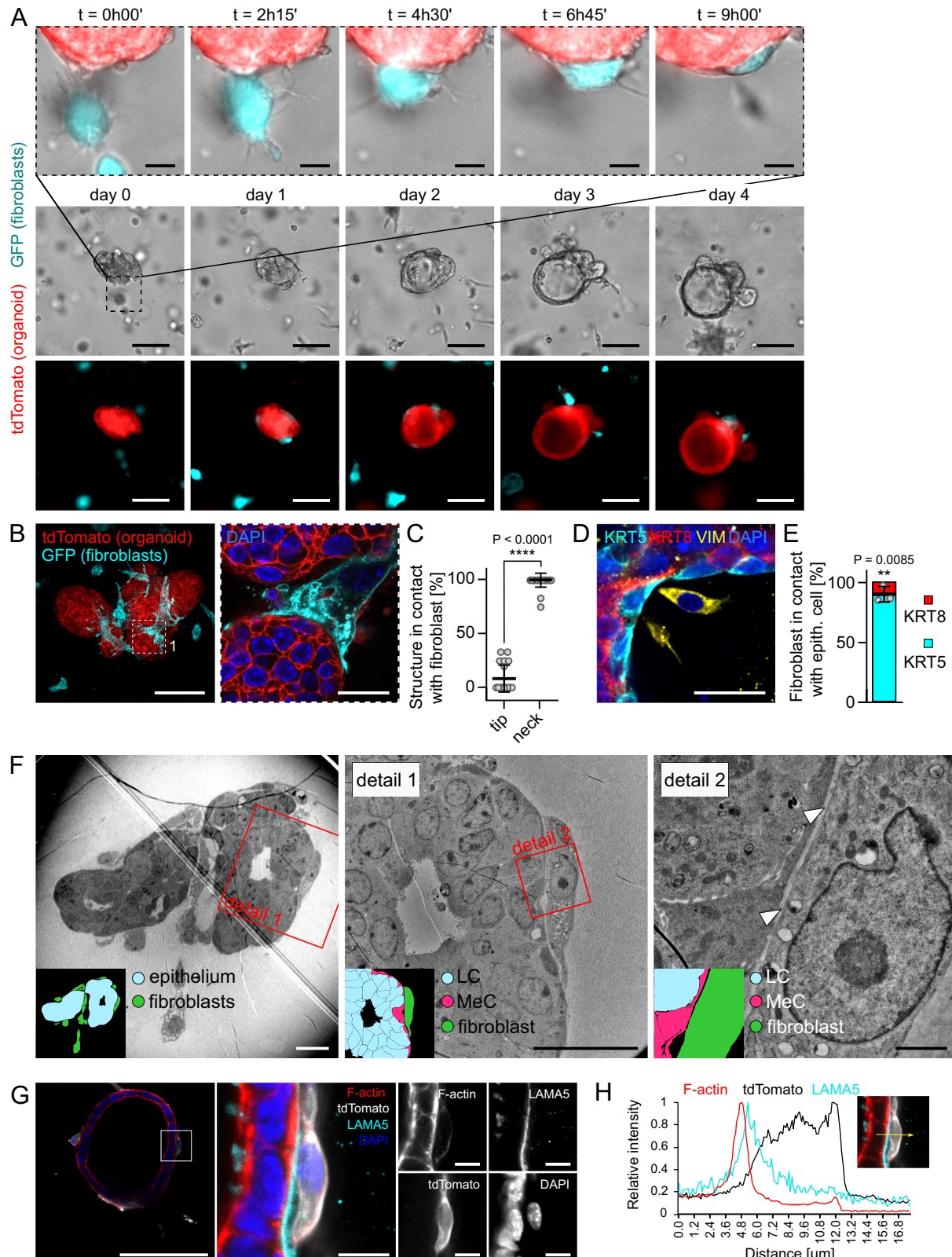

**Fig 3. Fibroblasts in co-cultures are in physical contact with the epithelium.** (A) Snapshots from time-lapse brightfield and fluorescence imaging of organoid (tdTomato) and fibroblast (GFP) co-culture (dispersed culture). Scale bar: 100 μm. Top line shows detail of fibroblast-organoid close interaction. Scale bar: 20 μm. (B, C) Images (B) and quantification (C) of the contact point between organoid (tdTomato) and

fibroblasts (GFP) on day 4 of co-culture (dispersed culture). Scale bar: 100 μm, scale bar in detail: 20 μm. **(C)** The plot shows mean ± SD, each dot represents 1 organoid, $n = 5$ experiments, $N = 21$ organoids. Statistical analysis: Two-tailored $t$ test. **(D)** Images of the contact point between organoid (luminal (KRT8) and myoepithelial (KRT5) cells) and fibroblasts (VIM) on day 5 of co-culture (dispersed culture). Scale bar: 20 μm. **(E)** Quantification of fibroblasts in contact with KRT5+ or KRT8+ epithelial cells. The plot shows mean ± SD, each dot represents average from 1 biological replicate, $n = 3$ experiments, $N = 14$ organoids, 219 fibroblasts. Statistical analysis: Two-tailored $t$ test. **(F)** Transmission electron micrographs and scheme (inset) of the contact point between luminal (LC, blue) and myoepithelial (MeC, magenta) cells and fibroblasts (green) on day 4 of co-culture (dispersed culture). Scale bar: 20 μm, scale bar in detail: 2 μm. In agreement with a published study (Ewald and colleagues), luminal cells are defined as lumen-facing cells, which present microvilli and numerous vesicles and granules. Myoepithelial cells are basally oriented, more elongated cells with less vesicles, granules, and organelles in the cytoplasm and they show a different electron density in their cytoplasm (it appears darker than the cytoplasm of luminal cells). The white arrowheads denote ECM between fibroblast and organoid. **(G)** Optical slice of organoid-fibroblast co-culture (dispersed culture), laminin 5 (cyan), DAPI (blue), F-actin (red), fibroblasts were isolated from *R26-mT/mG* mice (tdTomato, white). Scale bar: 100 μm, scale bar in detail: 10 μm. **(H)** A representative 1D relative fluorescence intensity plot. The measurement line is depicted in yellow (right). The data underlying the graphs shown in the figure can be found in S1 Data. ECM, extracellular matrix.

results demonstrate that fibroblast contractility is not necessary for fibroblast migration towards organoids but is required to induce organoid branching in co-cultures.

The need of fibroblast contractility for inducing epithelial branching suggests a mechanical signal transduction from fibroblasts to epithelium; therefore, we examined the subcellular localization of yes associated protein (YAP), a mechano-sensor that in a resting cell resides in the cytoplasm but translocates to the nucleus upon mechanical stress [24]. We found that YAP specifically accumulated in the nuclei of epithelial cells in the neck region of epithelial branch of the co-culture, the region in contact with contractile fibroblasts (Fig 4G and 4H). Importantly, this pattern was not present in branches induced with FGF2 (Fig 4G and 4I), indicating that YAP activation in epithelial cells at the necks of elongating buds is induced by the contact with contractile fibroblasts and not simply by the overall shape of the epithelial bud. However, knockout of *Myh9* in fibroblasts resulted in round organoids with no branching and a homogeneous distribution of nuclear YAP (Fig 4J). Our results show that while fibroblast contractility is necessary for the formation of branch with patterned YAP signal, the nuclear translocation of YAP can happen even in the absence of fibroblast contractility.

## Fibroblast-induced epithelial branching requires epithelial proliferation

Activation of YAP signaling followed by its nuclear translocation is often associated with cell proliferation [24]. To investigate if such association is manifested in the organoids, we performed EdU labeling for proliferative cells. In the co-cultures, we detected highest cell proliferation in the stalks of the branches (S10A–S10C Fig), i.e., in the areas of YAP nuclear localization (Fig 4G). In contrast, in the FGF2-treated organoids, no such pattern of EdU+ cells was observed (S10A, S10B and S10D Fig).

To test whether epithelial proliferation (and thus expansion) plays a role in organoid branching in co-cultures, we inhibited cell proliferation using aphidicolin (DNA polymerase inhibitor), upon which we observed a severe defect in organoid branching (S10E–S10G Fig). To test for the possibility that the observed effect could be caused by inhibition of fibroblast proliferation, we performed the experiment also with fibroblasts pretreated with mitomycin C, an irreversible DNA synthesis blocker (S10E Fig). The pretreatment of fibroblast with mitomycin C had no effect on the result (S10E–S10G Fig), demonstrating that fibroblast proliferation is dispensable while epithelial proliferation is necessary for organoid branching in co-cultures. In concordance with the results from organoid co-cultures, in the MCF7-ras spheroid co-culture model the blockage of spheroid proliferation by mitomycin C pretreatment of spheroids decreased spheroid size expansion and dramatically decreased branching of the MCF7-ras spheroids (S11A and S11B Fig).

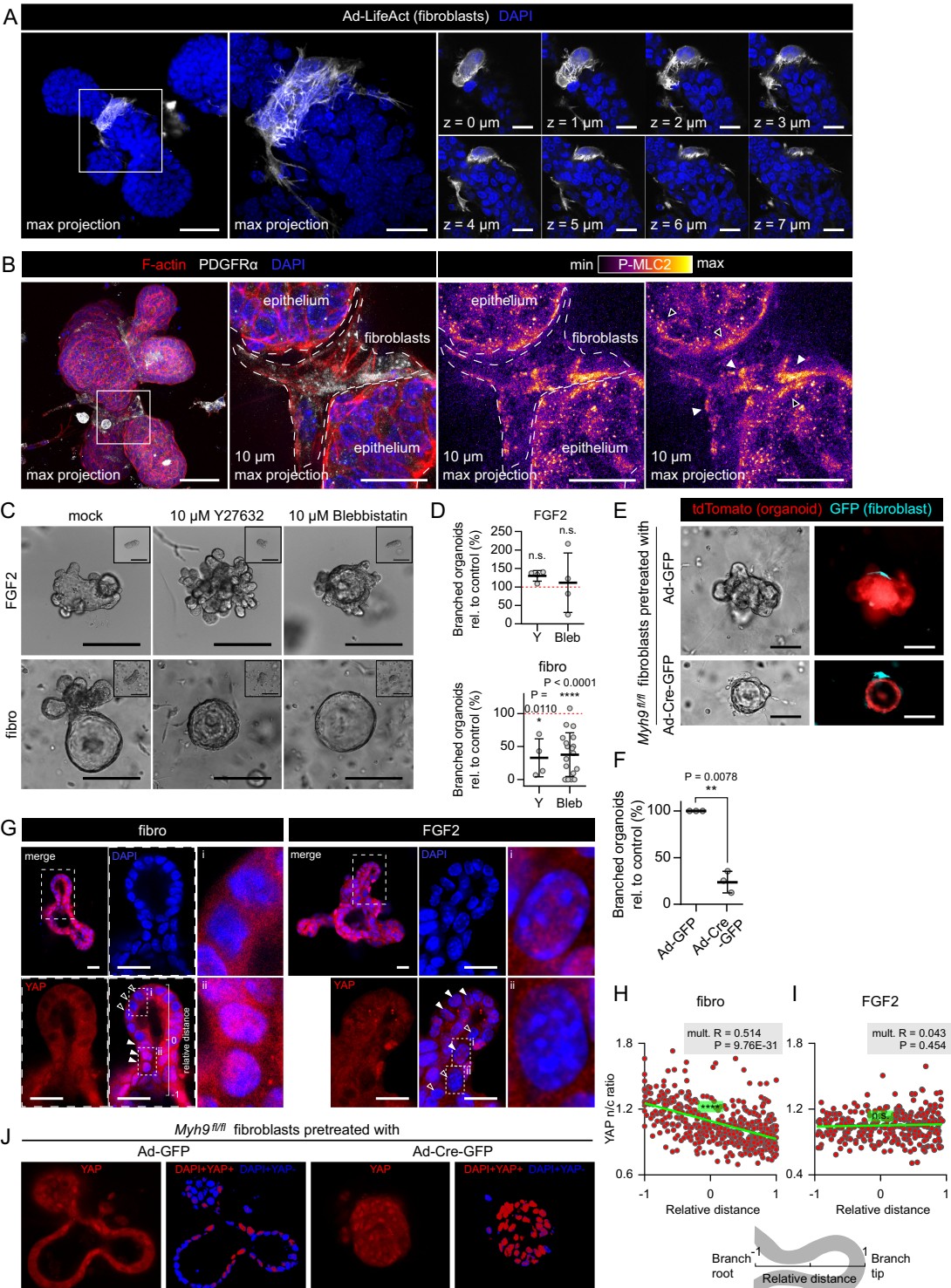

**Fig 4. Fibroblast-induced branching requires fibroblast contractility. (A)** Maximum projection (left), detailed maximum projection (middle), and detail optical sections (right) of organoid branch induced in co-culture with fibroblasts (day 4 of dispersed culture). DAPI (blue), fibroblasts prelabeled with Ad-LifeAct-GFP (white). Scale bar: 50 μm, scale bar in detail: 20 μm. **(B)** Full maximum projection (left) and 10 μm maximum projection (both middle and right) of organoid co-cultured with fibroblasts (dispersed culture). F-actin (red), PDGFRα (white), DAPI (blue), and phosphorylated myosin light chain 2 (P-MLC2, fire LUT). Scale bar: 50 μm, scale bar in detail: 20 μm. **(C)** Images of organoids on day 5 of culture (with day 0 insets) with FGF2 or

with fibroblasts (dispersed culture) and treated with mock (DMSO) or contraction inhibitors (ROCK1/2 inhibitor Y27632 or non-muscle myosin II inhibitor blebbistatin). Scale bar: 100 μm. **(D)** The plots show organoid branching with contraction inhibitors as mean ± SD. Statistical analysis: Multiple *t* tests between each treatment and the mock-treated control; *n* = 4–18 (each dot represents a biologically independent experiment), *N* = 20 organoids per experiment. **(E)** Images of tdTomato organoids on day 5 of co-culture with control or *Myh9* knock-out fibroblasts (dispersed culture). Scale bar: 100 μm. Videos from the 5-day experiment are presented in S5 Movie. **(F)** The plot shows organoid branching with control of *Myh9* knock-out fibroblasts from experiment in (**E**) as mean ± SD. Statistical analysis: two-tailored paired *t* test; *n* = 3 (each dot represents a biologically independent experiment), *N* = 20 organoids per experiment. **(G)** Staining of YAP in an organoid co-cultured with fibroblasts (dispersed culture) or with FGF2. Scale bar: 20 μm. Full arrowheads point to cells with nuclearly localized YAP, empty arrowheads point to cells with cytoplasmic YAP. **(H, I)** Quantification of YAP nuclear/cytoplasmic signal ratio. The scheme explains relative distance: −1 is branch root, +1 is branch tip. Each dot represents a single cell, *n* = 436 cells from 19 branches of 10 organoids (fibro, **H**); *n* = 306 cells from 12 branches of 10 organoids (FGF2, **I**). Statistical analysis: Linear regression, mult. R indicates correlation coefficient; P is the result of ANOVA F-test. **(J)** Staining of organoids co-cultured with control or *Myh9* knock-out fibroblasts. Scale bar: 200 μm. The data underlying the graphs shown in the figure can be found in S1 Data. FGF2, fibroblast growth factor 2; YAP, yes associated protein.

## Evidence for the role of fibroblast contractility in epithelial morphogenesis in vivo

During puberty, the period of major mammary epithelial growth and branching, new primary mammary epithelial branches arise through bifurcation of TEBs. Could contractile fibroblasts play a role in this process? TEBs are large and highly proliferative stratified epithelial structures consisting of multiple (5–10) layers of luminal cells (called body cells) and an outer layer of basal cells (or cap cells). Such structures are not replicated in our co-culture model because we model only a part of the complex in vivo microenvironment—the effect of fibroblasts. In vivo the TEBs are surrounded by a complex stroma, which provides instructions for epithelial morphogenesis, including besides fibroblasts several more stromal cell types (adipocytes, immune cells) that secrete paracrine signals for epithelial proliferation [25–30]. While our reductionist in vitro co-culture model (consisting of fibroblasts and epithelial organoids in basal medium with no exogenous morphogenetic growth factors) was essential for untangling the importance of the contact versus paracrine signaling in fibroblast-induced branching, to model TEBs we needed to modify our co-culture model to promote epithelial proliferation.

The classic mammary organoid model cultured in Matrigel with FGF2 [21] mimics epithelial stratification to some extent (reaching 3–4 layers of luminal cells in TEB-like ends of the branches) but does not support full myoepithelial coverage of branches [31]. We revealed that a stabilized form of FGF2 (FGF2-STAB; [32,33]) induces several TEB-like features in the organoids, including highly proliferative phenotype, multiple layers of luminal cells, and full myoepithelial cell coverage [34]. When we exposed the dispersed organoid-fibroblast co-cultures to FGF2-STAB, the organoids developed large branches with a set of features typical of TEBs in vivo, including stratified luminal cells, full myoepithelial coverage, and presence of basal-in-luminal cells (similar to cap-in-body cells in vivo [4]) (Fig 5A–5G). These results demonstrate that combination of contractile fibroblasts and strong proliferative signals can reproduce several aspects typical for TEB branching in organoid cultures (Fig 5H).

Finally, we sought to determine whether the contractility-dependent mechanism of fibroblast-induced branching could take place in vivo. We found fibroblasts expressing a contractility marker alpha smooth muscle actin (αSMA) in developing mammary glands during puberty (Fig 6A). The αSMA+ fibroblasts specifically populate the stroma surrounding the TEBs (Fig 6A and 6B), the actively growing part of epithelium, which produces new branches. Importantly, the fibroblasts in co-cultures do express αSMA as well (Fig 6C). To visualize the organization of fibroblasts in the peri-TEB stroma, we performed whole organ immunostaining, clearing and imaging of the mammary gland. We observed that fibroblasts (stained for their cytoskeletal marker vimentin) were organized perpendicularly to the nascent bud in

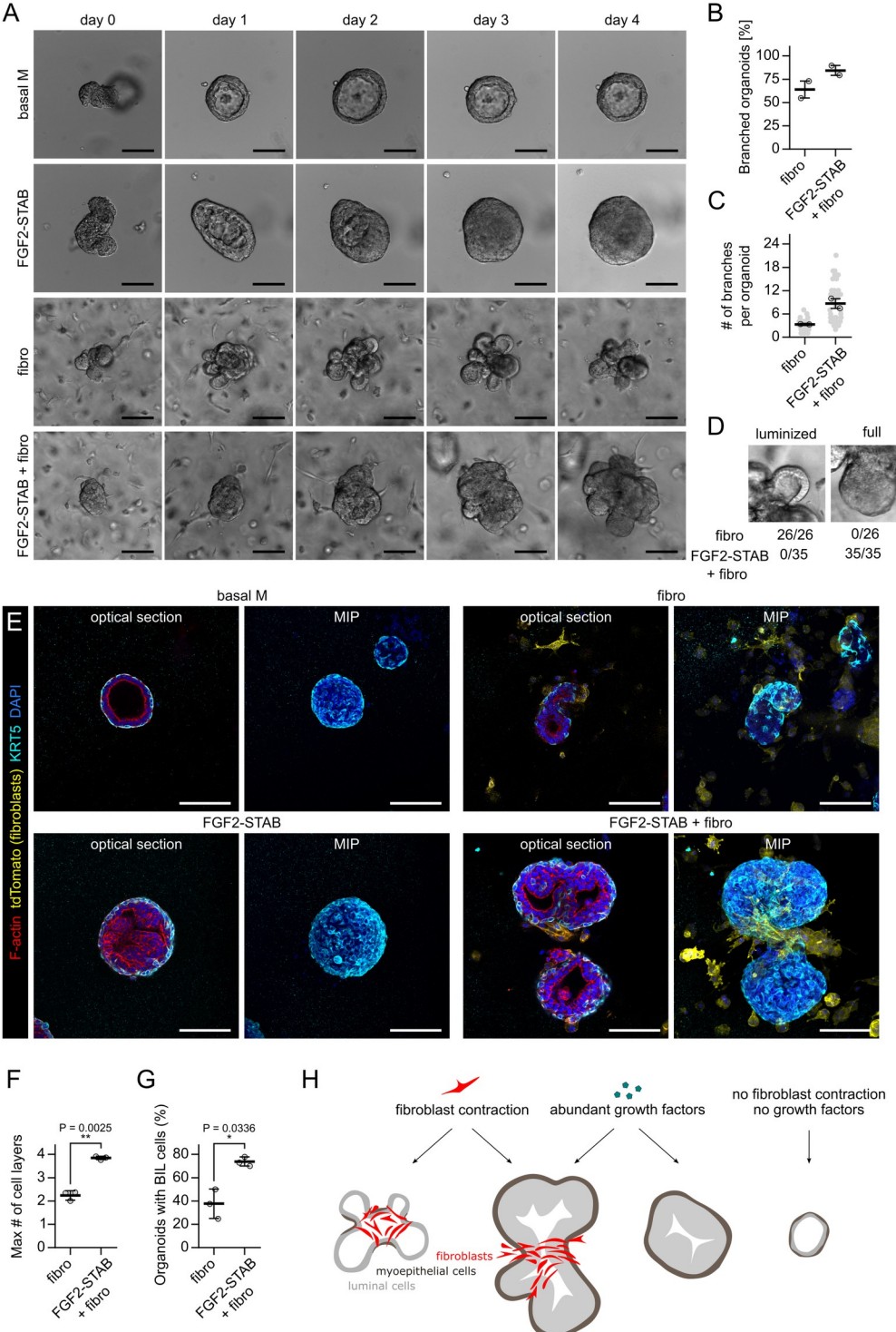

**Fig 5. Combination of fibroblasts and FGF2-STAB induces TEB-like phenotype of organoids. (A)** Time-lapse snap-shots of organoids grown in basal organoid medium with no exogenous growth factors (basal M), with FGF2-STAB, co-cultured with fibroblasts or co-cultured with fibroblasts with FGF2-STAB. Scale bar: 100 μm. **(B)** Quantification of organoid branching. The plot shows mean ± SD. *n* = 2 independent biological replicates, *N* = 20 organoids per experiment. **(C)** Quantification of number of branches per branched organoid. The plot shows mean ± SD. *n* = 2 independent biological replicates, *N* = 12–19 branching organoids per experiment. **(D)** Examples of luminized and full branch on bright-field imaging and quantification of the branch phenotypes. *n* = 2 independent biological replicates, *N* = 12–19 branching organoids per experiment. **(E)** Representative confocal images of organoids

on day 5 of culture with FGF2-STAB or fibroblasts. Scale bar: 100 μm. **(F)** Quantification of maximum number of cell layers in a branch in confocal images. The plot shows mean ± SD. The dots represent averages from individual experiments. Statistical analysis: two-tailored $t$ test; $n$ = 3 independent biological replicates, $N$ = 9–13 organoids per experiment. **(G)** Quantification of the percentage of organoids with KRT5+ cells present within the layers of KRT5-cells (basal-in-luminal, BIL cells) in confocal images. The plot shows mean ± SD. Statistical analysis: two-tailored $t$ test; $n$ = 3 independent biological replicates, $N$ = 9–13 organoids per experiment. **(H)** A schematic representation of uncoupling fibroblast contraction and growth factor signaling in organoids. The data underlying the graphs shown in the figure can be found in S1 Data. FGF2, fibroblast growth factor 2; TEB, terminal end bud.

bifurcating TEB (Fig 6D–i and S6 Movie) and perpendicularly to the epithelial growth direction at the TEB neck (Fig 6D-ii-1 and 6E and S7 Movie), forming loops similar to those observed in in vitro co-cultures (Figs 4A and 6F). On the other hand, fibroblasts surrounding subtending duct formed a less organized, mesh-like structure (Fig 6D-ii-2 and 6E). Together, our findings suggest that contractile fibroblasts could play a role in bifurcation of TEBs during branching morphogenesis in puberty.

## Discussion

Mechanical forces are an integral part and a driving factor of tissue morphogenesis. However, the sources of mechanical forces in different tissues are still unclear and little is understood of how force sensing is translated into cell fate during organ formation. Our work reveals the critical role of fibroblast-derived mechanical forces in regulation of mammary epithelial branching morphogenesis. It demonstrates that mammary fibroblasts generate mechanical forces via their actomyosin apparatus and transmit them to the epithelium, which leads to epithelial deformation and patterning of epithelial intracellular signaling, resulting in epithelial folding into branched structures.

### Fibroblast-generated mechanical forces as part of complex tissue mechanics

The role of intraepithelial forces in morphogenetic processes involving tissue folding, such as gastrulation, tubulogenesis, or buckling has been long recognized and intensively studied [35]. Similarly, the instructive role of mechanical properties and 3D organization of the ECM in determination of cell fate and behavior during organ formation, including mammary epithelial branching morphogenesis, has been well established [10,31,36]. However, the evidence for regulation of epithelial morphogenesis by mechanical stimuli from mesenchymal cells was discovered only recently and has been scarce, limited to the morphogenesis of feather buds in chick skin by mechanically active dermal cells [16], gut villification [17], and lung epithelial bifurcation and alveologenesis induced by smooth muscle cells or myofibroblasts [18–20,37].

### Fibroblasts as central regulators of epithelial morphogenesis and homeostasis: Evidence for mechanically active fibroblasts in vivo

Fibroblasts accompany mammary epithelial cells from early development through homeostasis to aging and disease and employ different functions to meet epithelial needs [15]. The multiple fibroblast functions are facilitated by fibroblast heterogeneity, which has only recently begun to be resolved using single-cell RNA sequencing approaches [38,39]. These studies confirmed well-established fibroblast roles in epithelial development and tissue homeostasis via production of paracrine signals and ECM, and fibroblast roles in regulation of immune landscape of the mammary gland. Though they did not detect mechanically active fibroblasts. However, these studies included only adult and aged mammary glands and omitted puberty, the stage of active mammary epithelial branching morphogenesis. Using immunostaining on mammary

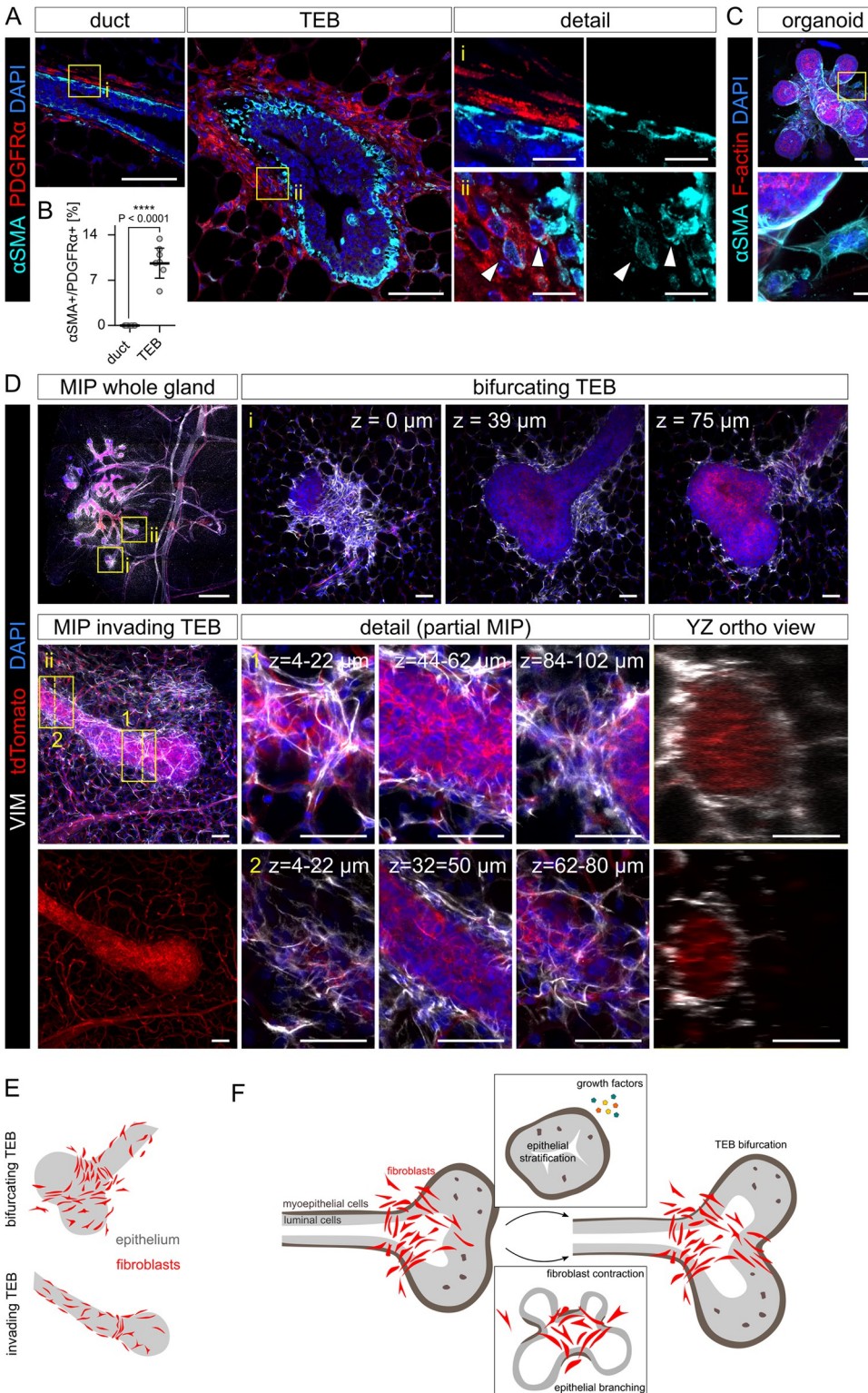

**Fig 6. Contractile fibroblasts surround mammary TEBs in vivo. (A)** PDGFRα and αSMA staining on pubertal mammary gland sections and detail of peri-ductal and peri-TEB fibroblasts. Scale bar: 50 μm and 10 μm in detail. **(B)** Quantification of αSMA+ cells out of PDGFRα+ fibroblasts. The plot shows mean ± SD, each dot represents 1 field of view, $n$ = 7 TEBs and 8 ducts; statistical analysis: $t$ test. **(C)** αSMA staining in a dispersed co-culture (day 5) and a detail of an αSMA+ fibroblast in contact with the organoid. Scale bar: 50 μm; scale bar in detail 10 μm. **(D)** Whole-mount

imaging of pubertal mammary gland stained for vimentin (VIM, a fibroblast marker). Detail (i) shows a bifurcating TEB. Detail (ii) shows an invading TEB, with close-up showing partial MIPs of upper, middle, and lower part of the TEB neck (1) and a subtending duct (2). The positions of the orthogonal YZ views are indicated with dashed yellow lines. Scale bar: 1 mm in the whole gland MIP, 50 μm in other images. **(E)** A schematic representation of fibroblasts surrounding TEBs in vivo and an organoid in vitro. Schemes were drawn from Figs 6D and 3B. **(F)** A schematic representation of our hypothesis on the role of contractile fibroblasts in TEB branching. Two mechanisms, paracrine signaling (growth factors) and mechanical cues (fibroblast contractility), which we uncoupled in vitro, work together to support mammary branching morphogenesis (TEB bifurcation) in vivo. The data underlying the graphs shown in the figure can be found in S1 Data. MIP, maximum intensity projection; TEB, terminal end bud.

glands in puberty, we discovered that mechanically active fibroblasts (contractile fibroblasts expressing αSMA, myofibroblasts) are localized specifically around TEBs, the main structures of epithelial branching during puberty, where they organize into structures similar to the loops observed in our fibroblast-organoid co-culture models. These data suggest that fibroblast contractility could play a role in mammary epithelial branching in vivo. Future studies employing myofibroblast-specific mouse models are needed to determine functional requirement of fibroblast contractility for mammary epithelial branching. Importantly, the presence and function of αSMA+ fibroblasts has been well documented in other developing or homeostatic organs, such as lung [40,41], intestine [42–44], or dermal sheath of the hair follicle [45,46]. While in the intestine the αSMA+ fibroblasts serve as a source of paracrine niche signals [42,44,47], the contractility of myofibroblasts is actively employed in alveolar septation [37] or relocation of the stem cell niche during hair cycle regression [45].

## The mechanism of fibroblast-induced mammary morphogenesis: Connection to ECM remodeling

It was previously proposed that mechanical forces generated by mesenchymal/stromal cells regulate epithelial morphogenesis indirectly via changes of ECM mechanics, including collagen I remodeling in embryonic gut [48] and postnatal mammary gland [5,14,15], or elastin deposition in the lung [40]. However, while not excluding contribution of such mechanism to mammary epithelial branching in vivo, our investigations in vitro in organoid-fibroblast cocultures devoid of collagen I demonstrate that collagen I fibers are not required for induction of epithelial folding by fibroblast contractility. Mammary fibroblasts form direct, highly dynamic contacts with mammary epithelial cells and induce a mechanosensitive response in the epithelium, resulting in patterning of a key morphogenetic regulator YAP. The direct contact between mammary fibroblasts and epithelial cells in vivo could be enabled by immature, thin basement membrane of TEBs [49], the highly proliferative epithelial structures, which drive pubertal mammary branching morphogenesis, and active remodeling of ECM by matrix metalloproteinases produced by both epithelial cells and fibroblasts [50], which is essential for mammary branching morphogenesis [50,51]. Our work does not exclude the importance of ECM remodeling by fibroblast mechanical forces in epithelial branching. We speculate that in vivo the highly dynamic mechanically active fibroblasts could initiate formation of epithelial clefts and further reinforce them by subsequent deposition and remodeling of ECM. Recently published simulations suggest that mammary pubertal branching is highly stochastic; however, its overall shape depends on the angle of TEB bifurcation [13,52]. Thus, the local effect of fibroblast contractility could affect the random branching pattern.

## The intimate relationship between fibroblasts and epithelium

Although our co-culture experiments demonstrate the need for direct contact between fibroblasts and epithelium for epithelial branching, paracrine interactions between the 2 are also

likely to be involved. Particularly at the beginning of the co-culture, as the fibroblasts migrate towards the organoid, they may follow epithelial signals such as PDGF or FGF ligands, which have been shown to be produced by epithelial cells [53,54] and to stimulate the directional migration of fibroblasts in vitro [7,53]. The degree of fibroblast motility in vivo and their directed migration to the epithelium during mammary development remains an open question, and the use of intravital imaging may shed light on this issue.

Direct molecular contact via heterotypic adhesions has been reported to promote cancer cell migration and invasion [55,56], but it has been less studied in normal epithelium. In our co-culture, we did not observe fibroblasts promoting invasive behavior in either normal organoids or cancer cell spheroids and we detected an ECM layer separating organoid from fibroblasts, suggesting that a different mode of cell–cell contact takes place. Moreover, the ability of fibroblasts to promote contact-dependent branching regardless whether in contact with myoepithelial cells of organoids or luminal cells of cancer spheroids suggest that fibroblasts could promote organoid branching without a proper molecular connection, just by forming a supracellular fibroblast structure that envelops the epithelium and applies contractile forces, as it was suggested for cancer-associated fibroblasts (CAFs) interacting with intestinal tumors [57].

## The mechanism of fibroblast-induced mammary morphogenesis: Requirement of paracrine signaling

Importantly, our results do not rule out the importance of fibroblast-secreted factors in mammary epithelial morphogenesis. However, we demonstrate that paracrine signals are not sufficient to drive organoid branching in the 3D in vitro cultures of organoids with fibroblasts without addition of any branching-inducing growth factors and show the importance of fibroblast-epithelium contact, so short-distance paracrine or juxtacrine signals could be important in the process. Several growth factors, including FGF2, FGF7, EGF, or TGFα can induce organoid branching in the absence of fibroblasts when added to the medium in nanomolar concentrations [58], including bifurcation of the organoid branches [59]. However, the evidence for requirement of those growth factors' expression in mammary fibroblasts for mammary epithelial branching in vivo is missing.

## The mechanism of fibroblast-induced mammary morphogenesis: Epithelial response

Our work reveals that the mechanical strain imposed on mammary epithelial cells by fibroblasts results in epithelial folding with negative curvature in the epithelial–fibroblast contact points. The part of epithelium with negative curvature, the stalk of the branch shows presence of epithelial cells with nuclear YAP and increased epithelial proliferation. In contrast, the organoids that branched in response to exogenous FGF2 did not show patterned cell proliferation or YAP nuclear localization, further accentuating different mechanisms underlying epithelial branching in response to growth factors and contractile fibroblasts. It remains unclear though whether fibroblasts induce YAP activation to promote epithelial proliferation at the neck and thus bud elongation, or if the patterned YAP activation in epithelial buds reflects the proliferative status of cells located in different regions of the organoid. Our data suggest that epithelial proliferation in the co-cultures is mechanistically linked to the contact with the fibroblasts and/or to the mechanical stress imposed by the contractile fibroblasts in the underlying epithelium and in its vicinity, possibly through juxtacrine signaling or mechanochemical interplay. Budding morphogenesis of stratified epithelium, such as in the FGF2-induced mammary organoids, might employ self-organizing mechanisms, including preferential cell-ECM adhesion versus cell–cell adhesion as demonstrated in salivary gland organoids [60]. We propose that in

vivo, in the complex microenvironment of the stroma-invading, growing and bifurcating TEB, it is the combined action of contractile fibroblasts and strong proliferative signals from the stroma that governs the morphogenetic process. It was demonstrated that although the whole TEB contains proliferative cells, it is the cells localized in the neck of the TEB, which will mainly contribute to the growth of the adjacent duct, not the cells localized in the TEB tip [61]. Because contractile fibroblasts surround specifically the neck region of the TEB, we speculate that they play an essential role in this process.

Importantly, the direct interactions between mammary epithelium (including both organoids from normal epithelium and spheroids from breast cancer cells) and fibroblasts do not lead to invasive dissemination of epithelial cells, unlike in co-cultures of squamous cell carcinoma with CAFs [55]. Interestingly, a recent study described mechanical compression of intestinal tumors by CAFs forming a mechanically active tumor capsule [57], providing further evidence for context-dependent employment of fibroblast-derived mechanical forces in tissue morphogenesis and tumorigenesis. In conclusion, we find that fibroblasts drive branching morphogenesis in mammary organoids by exerting mechanical forces on epithelial cells. These observations support the hypothesis that contractile fibroblasts drive pubertal mammary branching; however, future in vivo studies will be needed to formally demonstrate this. It is conceivable that such conserved mechanism could be used to regulate morphogenesis of other branched organs, providing a comprehensive understanding of overlapping but divergent employment of mechanically active fibroblasts in developmental versus tumorigenic programs.

## Materials and methods

### Animals

All procedures involving animals were performed under the approval of the Ministry of Education, Youth and Sports of the Czech Republic (license # MSMT-9232/2020-2), supervised by the Expert Committee for Laboratory Animal Welfare of the Faculty of Medicine, Masaryk University, at the Laboratory Animal Breeding and Experimental Facility of the Faculty of Medicine, Masaryk University (facility license #58013/2017-MZE-17214), or under the approval of the ethics committee of the Institut Curie and the French Ministry of Research (reference #34364–202112151422480) in the Animal Facility of Institut Curie (facility license #C75–05–18). ICR mice were obtained from the Laboratory Animal Breeding and Experimental Facility of the Faculty of Medicine, Masaryk University. *R26-mT/mG* [62] and *Acta2-CreERT2* mice [63] were acquired from the Jackson Laboratories. LifeAct-GFP mice [64] were created by Wedlich-Söldner team, *Myh9*$^{fl/fl}$ mice [65] were kindly provided by Dr. Sara Wickström. Transgenic animals were maintained on a C57/BL6 background. Experimental animals were obtained by breeding of the parental strains, the genotypes were determined by genotyping. The mice were housed in individually ventilated or open cages, all with ambient temperature of 22°C, a 12 h:12 h light:dark cycle, and food and water ad libitum. Female 4 to 8 weeks old virgin mice were used in the experiments. Mice were euthanized by cervical dislocation and mammary gland tissues were collected immediately.

### Primary mammary organoid and fibroblast isolation and culture

Primary mammary fibroblasts and organoids were isolated from 6 to 8 weeks old female virgin mice (ICR, unless otherwise specified) as previously described [66]. The mammary glands were chopped and partially digested in a solution of collagenase and trypsin [2 mg/ml collagenase A, 2 mg/ml trypsin, 5 µg/ml insulin, 50 µg/ml gentamicin (all Merck), 5% fetal bovine serum (FBS; Hyclone/GE Healthcare) in DMEM/F12 (Thermo Fisher Scientific)] for 30 min

at 37°C. Resulting tissue suspension was treated with DNase I (20 U/ml; Merck) and submitted to 5 rounds of differential centrifugation (450 × g for 10 s) to separate epithelial (organoid) and stromal fractions. The organoids were resuspended in basal organoid medium [1 × ITS (10 μg/ml insulin, 5.5 μg/ml transferrin, 6.7 ng/ml sodium selenite), 100 U/ml of penicillin, and 100 μg/ml of streptomycin in DMEM/F12] and kept on ice until co-culture setup. The cells from the stromal fraction were pelleted by centrifugation, suspended in fibroblast cultivation medium (10% FBS, 1× ITS, 100 U/ml of penicillin, and 100 μg/ml of streptomycin in DMEM) and incubated on cell culture dishes at 37°C, 5% $CO_2$ for 30 min. Afterwards, the unattached (non-fibroblast) cells were washed away, the cell culture dishes were washed with PBS and fresh fibroblast medium was provided for the cells. The cells were cultured until about 80% confluence. During the first cell subculture by trypsinization, a second round of selection by differential attachment was performed, when the cells were allowed to attach only for 15 min at 37°C and 5% $CO_2$. The fibroblasts were expanded and used for the experiments until passage 5.

To inhibit fibroblast proliferation for specific assays, the fibroblasts were treated with 10 μg/ml mitomycin C in fibroblast medium for 3 h at 37°C, 5% $CO_2$. Afterwards, the fibroblasts were washed 3 times with PBS and 1 time with basal organoid medium, trypsinized and used to set up co-cultures.

To prepare fibroblast-conditioned medium, the fibroblasts were seeded in cell culture dishes in fibroblast medium and the next day, the cells were washed 3 times with PBS and incubated with basal organoid medium for 24 h. Afterwards, the medium was collected from the dishes, sterile-filtered through a 0.22 μm filter, and used immediately in the experiment, or aliquoted, stored at −20°C and used within 5 days of conditioned medium preparation.

## 3D culture of mammary organoids and fibroblasts

3D culture of mammary organoids and fibroblasts was performed as previously described [67]. Freshly isolated mammary organoids were embedded in Matrigel either alone (300 organoids in 45 μl of Matrigel per well) or with $5 \times 10^4$ mammary fibroblasts per well and plated in domes in 24-well plates. For transwell experiments, organoids were plated in domes in the transwell (8 μm pore size, Falcon-Corning), fibroblasts were plated in lower chamber. After setting the gel for 45 to 60 min at 37°C, the cultures were overlaid with basal organoid medium (1× ITS, 100 U/ml of penicillin, and 100 μg/ml of streptomycin in DMEM/F12), not supplemented or supplemented with growth factors [2.5 nM FGF2 (Enantis) or 2.5 nM FGF2-STAB (Enantis)] or small molecule inhibitors (S1 Table) according to the experiment. The cultures were incubated in humidified atmosphere of 5% $CO_2$ at 37°C on Olympus IX81 microscope equipped with Hamamatsu camera and CellR system for time-lapse imaging. The organoids/co-cultures were photographed every 60 min for 5 days with manual refocusing every day (high-detail imaging) or photographed only once per day for 5 days (low-detail imaging). The images were exported and analyzed using Image J. Organoid branching and retraction was evaluated from videos and it was defined as formation (or loss) of a new bud/branch from the organoid. Organoids that fused with another organoid or collapsed after attachment to the bottom of the dish were excluded from the quantification. Quantification of fibroblast-organoid contacts was performed manually in ImageJ. Quantification of branch thickness was performed on images from day 5 of culture, manually in ImageJ.

For fluorescent time-lapse imaging, organoids were isolated from *R26-mT/mG* mammary glands on day of the experiment. Fibroblasts were isolated from *Acta2-CreERT2;mT/mG* mice, cultured to passage 2–3 and induced in vitro by 0.5 mM 4-OH-tamoxifen (Sigma) treatment for 4 days prior to trypsinization and experimental use. Before experiment, the GFP

fluorescence of fibroblasts was assessed using a microscope and when it was > 95%, the cells were used for co-culture. Co-cultures were seeded on coverslip-bottom 24-well plate (IBIDI) and imaged on Cell Discoverer 7 equipped with PLAN-APOCHROMAT 20×/0.95 autocorr with 0.5× magnification lens. GFP was imaged with 470/40 nm excitation, 525/50 nm emission, tdTomato was imaged with 545/25 nm excitation, 605/70 nm emission filter (all Zeiss). The samples were incubated in a humidified atmosphere of 5% $CO_2$ at 37˚C during the imaging.

## 3D culture of spheroids and fibroblasts

MCF7-ras cells ([68] kindly provided by Dr. Ula Polanska) were expanded in DMEM/F12 supplemented with 10% FBS, 100 U/ml of penicillin, and 100 μg/ml of streptomycin and incubated in non-adherent PolyHEMA-coated dish overnight to form spheroids. Next day, the spheroids were embedded either alone (200 spheroids in 45 μl of Matrigel per well) or with $5 \times 10^4$ mammary fibroblasts per well and plated in domes in 24-well plates. After setting the gel for 45 to 60 min at 37˚C, the cultures were overlaid with basal organoid medium, supplemented with growth factors [2.5 nM FGF2 (Enantis) or EGF (Peprotech)] small molecule inhibitors (S1 Table) according to the experiment. The (co-)cultures were incubated in a humidified atmosphere of 5% $CO_2$ at 37˚C on Olympus IX81 microscope equipped with Hamamatsu camera and CellR system for time-lapse imaging and photographed every 60 min for 5 days with manual refocusing every day (high-detail imaging) or photographed only once per day for 5 days (low-detail imaging). The images were exported and analyzed using Image J. Spheroid budding was evaluated from the videos and it was defined as formation of a new bud from the spheroid. Spheroids that fused with other spheroids were excluded from the quantification.

## Knockdown and knockout of *Myh9* in mammary fibroblasts

For *Myh9* knockdown, the pre-designed Silencer Select siRNAs against *Myh9* (IDs s70267 and s70268, Myh9si#1 and Myh9si#2, respectively) and the scrambled negative control siRNA (Silencer Select negative control or Stealth negative control siRNA), all from Thermo Fisher Scientific, were transfected into wild-type (ICR) fibroblasts with Lipofectamine 3000 Reagent (Thermo Fisher Scientific) according to manufacturer's instructions at 20 nM siRNA. For *Myh9* knockout, *Myh9^{fl/fl}* fibroblasts were transduced with adenoviruses Adeno-Cre-GFP (Ad-Cre-GFP) or Adeno-GFP (Ad-GFP) from Vector Biolabs at 200 MOI for 4 h. Next day, the transfected/transduced fibroblasts were put in co-culture with organoids and submitted to bright-field or fluorescent time-lapse imaging. A part of the fibroblasts was further cultured and knockdown/knockout efficiency was determined 72 h after transfection/transduction by qPCR analysis of *Myh9* mRNA levels, normalized to housekeeping genes *Actb* and *Eef1g*, and by immunostaining for MYH9.

## LifeAct adenoviral transduction

For imaging experiments with LifeAct, fibroblasts were infected with LifeAct adenoviral particles (IBIDI) according to the manufacturer's instructions prior to co-culture set-up. Briefly, the adenovirus particles were reconstituted in fibroblast cultivation medium at concentration of 500 MOI and incubated with adherent fibroblasts at 37˚C for 4 h. After that, adenovirus-containing medium was washed out, and the cells were kept overnight in fibroblast cultivation medium. The next day, GFP fluorescence was checked under the microscope and when >50% of cells appeared green, fibroblasts were used for co-culture.

## Immunofluorescence staining of 2D fibroblasts

For immunofluorescent analysis, fibroblasts were cultured directly on glass coverslips, fixed with 10% neutral buffered formalin, permeabilized with 0.05% Triton X-100 in PBS and blocked with PBS with 10% FBS. Then, the coverslips were incubated with primary antibodies (S2 Table) for 2 h at RT or overnight at 4˚C. After washing, the coverslips were incubated with secondary antibodies and phalloidin AlexaFluor 488 (S2 Table) for 2 h at RT. Then, the coverslips were washed, stained with DAPI (1 μg/ml; Merck) for 10 min and mounted in Mowiol (Merck). The cells were photographed using Axio Observer Z1 microscope with laser scanning confocal unit LSM 800 with 405, 488, 561, and 640 nm lasers, GaAsp PMT detector and objective Plan-Apochromat 40×/1.20 and C-Apochromat 63× /1.20 with water immersion (all Zeiss). The brightness of each channel was linearly enhanced in Zen Blue software (Zeiss) and pictures were cropped to final size in Photo Studio 18 (Zoner).

## Immunofluorescence staining of 3D co-cultures

For immunofluorescent analysis of 3D co-cultures, the co-cultures were fixed with 10% neutral buffered formalin, washed, and stored in PBS. Next, organoid co-cultures were stained according to the droplet-based method as described [69]. Briefly, the fixed co-cultures were placed on stereoscope (Leica FM165C) and pieces containing an organoid with approximately 100 μm of surrounding Matrigel with fibroblasts were manually cut out with 25G needles and moved on parafilm-covered cell culture dish for staining. All the staining steps were done on the parafilm in 20 μl drops, and all solutions were changed under direct visual control using the stereoscope. The co-cultures were permeabilized with 0.5% Triton X-100 in PBS, blocked with 8% FBS and 0.1% Triton X-100 in PBS (3D staining buffer, 3SB) and incubated with primary antibodies (S2 Table) in 3SB over 1 to 3 nights at 4˚C. Then, the co-cultures were washed for 3 h with 0.05% Tween-20 in PBS and incubated with secondary antibodies, phalloidin AlexaFluor 488 (S2 Table) and DAPI (1 μg/ml; Merck) in 3SB over 1 to 2 nights at 4˚C in dark. Then, the co-cultures were washed for 3 h with 0.05% Tween-20 in PBS, cleared with 60% glycerol and 2.5 M fructose solution overnight at RT in dark and mounted between slide and coverslip with double-sided tape as a spacer. The co-cultures were imaged using inverted microscope Axio Observer 7 with laser scanning confocal unit LSM 880 with 405, 488, 561, and 633 nm lasers, GaAsp PMT spectral detector and objective C-Apochromat 40×/1.20 or C-Apochromat 63×/1.20 with water immersion (all Zeiss). The co-cultures were photographed either as one optical slice or as 3D z-stacks of various z-step as required per experiment. The brightness of each channel was linearly enhanced in Zen Blue software (Zeiss) and pictures were cropped to final size in Photo Studio 18 (Zoner). Image analysis was done manually in ImageJ. Contact of fibroblasts and organoids in confocal images was analyzed in organoid middle section by measuring the perimeter of the organoid in contact with PDGFRα signal and without it. Fibroblast loop was defined as crescent shaped tdTomato signal that wrapped at least half of organoid branch. EdU signal was quantified in 3 to 5 optical sections of organoid 20 μm apart to avoid multiple counts from the same cell. Contact between fibroblast and KRT5/KRT8 cells, number of basal-in-luminal cells was counted manually in ImageJ. Number of cell layers in organoids was counted manually in the thickest part of a branch. LAMA5 signal along a line was measured in ImageJ.

## EdU incorporation assay

For proliferation analysis, 5-ethynyl-2′-deoxyuridine (EdU) incorporation click-it kit (Thermo Fisher Scientific) was used. EdU was administered to the organoid co-cultures 2 h prior to fixation and the EdU signal was developed according to the manufacturer's instructions prior to immunofluorescence staining. The volumes were adjusted for the droplet-based staining as above.

## Immunohistochemistry-immunofluorescence (IHC-IF)

Mammary glands #4 were harvested from 6 weeks old females, fixed in 10% neutral buffered formalin overnight, dehydrated in ethanol with increasing concentration and xylene and embedded in paraffin, and 5 μm sections were cut on rotational microtome (Thermo Scientific, Microm HM340E). After rehydration, sections were boiled in pH9 Tris-EDTA buffer to retrieve antigens, blocked in 10% FBS and incubated with primary and secondary antibodies, mounted (Aqua Poly/Mount, Polysciences) and imaged on laser scanning confocal microscope (LSM780/880, Zeiss). The quantification of PDGFRα and αSMA positive cells was done manually in ImageJ, considering DAPI signal to distinguish single cells and continuous αSMA signal as a border of epithelium. The fields of view were scored "duct" or "TEB" based on morphology of the structures (TEBs: stratified epithelium, bulb-like shape, presence of cap-in-body cells, cuboidal cap cells; duct: one layer of luminal cells, elongated myoepithelial cells) and on the position of the structure at the distal part of the mammary epithelium (invasive front).

## Immunofluorescence staining of whole-mount cleared mammary gland

Staining and clearing of mammary glands was done following clear, unobstructed brain imaging cocktails (CUBIC) protocol [70,71]. Briefly, mammary glands #3 of 4 weeks old females were harvested and fixed in 10% neutral buffered formalin overnight, washed and incubated in CUBIC reagent 1 (25% (w/w) urea, 25% (w/w) N,N,N',N'-tetrakis(2-hydroxypropyl)ethylenediamine, 15% (w/w) Triton X-100 in distilled water) for 4 days shaking at RT. After washing, the glands were blocked using blocking buffer (5% FBS, 2% BSA, 1% Triton X-100, 0.02% sodium azide in PBS) overnight at RT, incubated with primary antibodies diluted in blocking buffer for 3 days at RT with rocking, washed 3 times for 2 h (0.05% Tween 20 in PBS) and incubated with secondary antibodies and DAPI (1 μg/ml) in blocking buffer. Then, the glands were transferred to CUBIC reagent 2 (50% (w/w) sucrose, 25% (w/w) urea, 10% (w/w) 2,2',2"-nitrilotriethanol, 0.1% (w/w) Triton X-100 in distilled water) for 2 days at RT with rocking. The samples were mounted with CUBIC reagent 2 between 2 coverslips with double-sided tape as a spacer to enable imaging from both sides and they were imaged on laser scanning confocal microscope LSM780 (Zeiss).

## Image analysis of signal distribution

The analysis of YAP nuclear to cytoplasmic ratio was done in ImageJ (NIH). Cells in optical section in the middle of an organoid branch were manually annotated and segmented for target protein signal (YAP channel) and nuclei (DAPI channel) and density of pixels in YAP channel in the regions of interest (ROIs) was measured. The nuclear to cytoplasmic ratio of YAP was calculated in Excel (Microsoft). The spatial information of each ROI was manually measured on a line parallel to the branch longitudinal axis and normalized, with the value "1" set for the tip of the branch and the value "−1" set for the root of the branch. The graphs and linear regression line were created in Prism 6 (GraphPad) or Excel. Colocalization analysis of YAP and DAPI channels was done in Zen Black (Zeiss) and presented as color-coded (blue DAPI+YAP- and red DAPI+YAP+). The same cut-off for the colocalization analysis was applied for all images from the same experiment.

## Real-time quantitative PCR (qPCR)

RNA from fibroblasts was isolated using RNeasy Mini Kit (Qiagen) according to the manufacturer's instruction. RNA concentration was measured using NanoDrop 2000 (Thermo Fisher Scientific). RNA was transcribed into cDNA by using Transcriptor First Strand cDNA

Synthesis Kit (Roche) or TaqMan Reverse Transcription kit (Life Technologies). Real-time qPCR was performed using 5 ng cDNA, 5 pmol of the forward and reverse gene-specific primers each (primer sequences are shown in S3 Table) in Light Cycler SYBR Green I Master mix (Roche) on LightCycler 480 II (Roche). Relative gene expression was calculated using the ΔΔCt method and normalization to 2 housekeeping genes, β-actin (*Actb*) and eukaryotic elongation factor 1 γ (*Eef1g*).

### Transmission electron microscopy

The 3D co-cultures were fixed with 3% glutaraldehyde in 100 mM sodium cacodylate buffer, pH 7.4 for 45 min, postfixed in 1% $OsO_4$ for 50 min, and washed with cacodylate buffer. After embedding in 1% agar blocks, the samples were dehydrated in increasing ethanol series (50, 70, 96, and 100%), treated with 100% acetone, and embedded in Durcupan resin (Merck). Ultrathin sections were prepared using LKB 8802A Ultramicrotome, stained with uranyl acetate and Reynold's lead citrate (Merck), and examined with FEI Morgagni 286(D) transmission electron microscope. The cells in the schematics were segmented manually.

### Statistical analysis

Sample size was not determined a priori and investigators were not blinded to experimental conditions. Statistical analysis was performed using GraphPad Prism software. Student's *t* test (unpaired, two-tailed) was used for comparison of 2 groups. Bar plots were generated by Graph-Pad Prism and show mean ± standard deviation (SD). *$P < 0.05$, **$P < 0.01$, ***$P < 0.001$, ****$P < 0.0001$. The number of independent biological replicates is indicated as *n*.

### Limitations of the study

The critical experiment that demonstrates the need of fibroblasts' physical contact with the epithelium for epithelial branching does not allow to distinguish between direct physical contact and potential juxtacrine or very short-distance paracrine signaling between the epithelium and fibroblasts, which may contribute to epithelial morphogenesis.

### Supporting information

**S1 Fig. MCF7-ras spheroids do not respond to exogenous growth factors by branching. (A)** Time-lapse snapshots of MCF7-ras spheroids cultured in basal medium with no exogenous growth factors (basal M) or with FGF2 or EGF. Scale bar: 100 μm. **(B)** Time-lapse snapshots of MCF7-ras spheroids co-cultured with no stromal cells (basal M) or with fibroblasts (fibro). Scale bar: 100 μm.
(TIFF)

**S2 Fig. Fibroblast-organoid contacts precede organoid branching. (A)** Time-lapse snapshots of an organoid-fibroblast co-culture. Scale bar: 100 μm. **(B)** Detailed snapshots of 3 examples of fibroblast-organoid contact establishment in the co-cultures shown in **(A)** on days 1, 2, and 3. Red arrowheads indicate fibroblasts of interest. Scale bar: 50 μm. **(C)** Quantification of organoid circularity (data from **Fig 1**), number of new branches and number of established fibroblast-organoid contacts from matched experiments. The plot shows mean ± SD; *n* = 3 (each dot represents the average from a biologically independent experiment, *N* = 20 organoids per experiment). **(D)** Maximum intensity projection (MIP) and optical section images of a dispersed co-culture on day 2.5, representative images of cystic and budding organoids (tdTomato). Fibroblasts were detected by immunostaining for PDGFRα. Scale bar: 100 μm. **(E)** Quantification of organoid middle section perimeter in contact with PDGFRα signal. The plot

shows mean ± SD. Each dot represents an average from 1 experiment. Statistical analysis: two-tailored *t* test; *n* = 3 independent biological samples, *N* = 15–24 organoids per sample. The data underlying the graphs shown in the figure can be found in S1 Data.
(TIFF)

**S3 Fig. Quantification of fibroblast loops.** **(A)** A representative confocal image of a dispersed co-culture on day 4. Scale bar: 20 μm, scale bar in detail: 10 μm. **(B)** A representative confocal image of a dispersed organoid-fibroblast co-culture on day 3. The arrowhead indicates the fibroblast loop at the branch neck. Scale bar: 50 μm. **(C)** Quantification of the presence of fibroblast loops around organoid branches in dispersed co-cultures. The plot shows mean ± SD. Statistical analysis: two-tailored *t* test; *n* = 3 independent biological replicates, *N* = 5–12 organoids per experiment; 56 branches in total. The data underlying the graphs shown in the figure can be found in S1 Data.
(TIFF)

**S4 Fig. MCF7-ras spheroid budding in co-cultures requires cell contractility.** **(A, C)** Photographs of spheroids on day 4 of dispersed co-culture with fibroblasts upon treatment with no inhibitor (mock), with blebbistatin (Bleb, **A**) or with Y27632 (**C**). Top gray and red bars indicate proportion of branched spheroids out of all spheroids per condition. Scale bar: 100 μm. **(B, D)** Quantification of number of branches/buds per branched spheroid in conditions from **(A)**. The plot shows mean ± SD, each lined dot shows mean from each experiment, each faint dot shows single spheroid measurement, *n* = 4 (**B**) or 5 (**D**) biologically independent experiments, *N* = 20 spheroids per experiment. Statistical analysis: two-tailored *t* test. The data underlying the graphs shown in the figure can be found in S1 Data.
(TIFF)

**S5 Fig. Contractility inhibitors do not impede fibroblast motility.** **(A)** Representative endpoint images of organoids in dispersed co-cultures with contractility inhibitors. Scale bar: 100 μm. **(B)** Detailed time-lapse snapshots of fibroblast-organoid contact establishment in co-cultures with or without the inhibitors. Scale bar: 50 μm. White arrowhead indicates the fibroblast of interest. **(C, D)** Quantification of fibroblast-organoid contacts established in co-cultures with inhibitors (Y = 10 μM Y27632, **C**; Bleb = 10 μM Blebbistatin, **D**) within the first 2 days. The plots show mean ± SD. Statistical analysis: two-tailored *t* test; *n* = 3 independent biological replicates, *N* = 10 organoids per experiment. The data underlying the graphs shown in the figure can be found in S1 Data.
(TIFF)

**S6 Fig. Fibroblast contractility is necessary for branch maintenance.** **(A)** Experimental scheme (top) and time-lapse snapshots of dispersed co-cultures treated with contractility inhibitors on day 3 of culture. Scale bar: 100 μm. White arrowheads indicate organoid branches. **(B–D)** Quantification of organoids with retracted branches (**B**), number of formed branches per branched organoids (**C**) and number of retracted branches per organoid (**D**). The plots show mean ± SD. Statistical analysis: two-tailored *t* test; *n* = 4 independent biological replicates, *N* = 20 organoids per experiment. The data underlying the graphs shown in the figure can be found in S1 Data.
(TIFF)

**S7 Fig. Mammary fibroblasts express MYH9 and MYH10.** **(A)** Real-time qPCR analysis of non-muscle myosin II heavy chain genes *Myh9*, *Myh10*, and *Myh14* in mammary fibroblasts (fib) and epithelium (organoids, org). Plots show mean ± SD. Statistical analysis: two-tailored *t* test; *n* = 3 independent biological samples. **(B)** Representative images of MYH9 and MYH10

immunostaining in mammary fibroblasts in the first passage. Scale bar: 50 μm. The data underlying the graphs shown in the figure can be found in S1 Data.
(TIFF)

**S8 Fig. Knockdown of *Myh9* in mammary fibroblasts abrogates fibroblast-induced branching of mammary organoids. (A, B)** Representative images (day 5 of culture) **(A)** and quantification **(B)** of organoid branching in dispersed co-cultures with wild-type fibroblasts pre-treated with nonsense (siNC) or *Myh9* targeting (siMyh9) siRNA. Plot indicates mean ± SD. Statistical analysis: two-tailored paired *t* test; $n = 6$ independent *Myh9* knockdown experiments; $N = 20$ organoids per each treatment of each independent experiment. Videos from the 5-day experiment are presented in S4 Movie. **(C–E)** Quantification of MYH9 down-regulation in *Myh9* KO fibroblasts by immunofluorescence. The plot **(C)** shows mean ± SD, $n = 2$ independent experiments. Representative images **(D)** show MYH9 (cyan) and F-actin (phalloidin, magenta) staining in cultured primary mammary fibroblasts from *Myh9*$^{fl/fl}$ mice, treated with adeno-GFP (Ad-GFP) or adeno-Cre-GFP (Ad-Cre-GFP) vector, including details **(E)** of cytoskeleton organization. Scale bars: 1 mm **(D)**, 20 μm **(E**, first and third row), and 5 μm **(E**, second and fourth row). The data underlying the graphs shown in the figure can be found in S1 Data.
(TIFF)

**S9 Fig. *Myh9* knock-out does not impede fibroblast motility. (A)** Detailed time-lapse snapshots of fibroblast-organoid contact establishment in dispersed co-cultures with control or *Myh9*-KO fibroblasts and tdTomato+ organoids. Scale bar: 50 μm. **(B)** Quantification of fibroblast-organoid contacts established in the first 3 days of co-culture, comparing GFP+ and GFP- fibroblasts (GFP is a marker of adenoviral transduction). The plot shows mean ± SD. Statistical analysis: two-tailored *t* test; $n = 3$ independent biological replicates, $N = 20$ organoids per experiment. The data underlying the graphs shown in the figure can be found in S1 Data.
(TIFF)

**S10 Fig. Proliferation in co-culture system. (A)** Representative images of organoids on day 4 of culture in basal medium (basal M), in dispersed co-culture with fibroblasts or with FGF2, EdU administered 2 h pre-fix, EPCAM (red), DAPI (blue), EdU (cyan), fibroblasts were isolated from *R26-mT/mG* mice (tdTomato, white). Scale bar: 100 μm. **(B)** Optical section of a branch from **(A)** (top), a scheme of branch regions (bottom). **(C, D)** Quantification of percentage of EdU+ cells from epithelial cells in different branch regions in fibroblast-organoid dispersed co-culture **(C)** and in FGF2-treated organoid **(D)**. The box and whiskers plot shows minimum, median, and maximum values, and second and third quartiles of data distribution. $n = 3$ independent experiments, $N = 6$ organoids, 2,202 analyzed cells in **(C)**; $N = 11$ organoids, 3,104 analyzed cells in **(D)**. Statistical analysis: Multiple *t* tests. **(E)** A scheme of the proliferation-inhibition experiment. **(F)** Co-cultures at day 5 (dispersed culture), fibroblasts pretreated with +/- mitomycin C (MMC), co-cultures treated with +/- aphidicolin (Aph). Scale bar: 100 μm. **(G)** Quantification of the percentage of branched organoids from experiment in **(F)**. The plot shows mean ± SD, each dot represents a biologically independent experiment, $n = 2$, $N = 51–77$ organoids per sample, statistical analysis: *t* test. The data underlying the graphs shown in the figure can be found in S1 Data.
(TIFF)

**S11 Fig. Spheroid proliferation is necessary for its branching in co-culture with fibroblasts. (A)** Representative images of MCF7-ras spheroids in dispersed co-culture with fibroblasts on day 4 with spheroids formed from mock- or mitomycin C-treated MCF7-ras cells. The insets

(top red bars) show proportion of branched spheroids out of all spheroids per condition. Scale bar: 100 μm. **(B)** The plot shows number of spheroid branches/buds formed, with mean ± SD. Each lined dot represents mean of each experiment, each faint dot represents 1 spheroid, *n* = 4 independent experiments (coded by dot colors), *N* = 15–20 spheroids per experiment. Statistical analysis: two-tailored *t* test. The data underlying the graphs shown in the figure can be found in S1 Data.
(TIFF)

**S1 Movie. Mammary epithelial branching morphogenesis upon FGF2 treatment or fibroblast co-culture.** The video is composed of time-lapse videos capturing 5 days of epithelial morphogenesis in 3D organoid culture with no growth factor in the basal organoid medium (left), with FGF2 in the basal organoid medium (middle), or in fibroblast-organoid co-culture without addition of any growth factors to the basal organoid medium. Snapshots from the videos are depicted in Fig 1A.
(AVI)

**S2 Movie. Fibroblasts dynamically interact with the epithelium.** Time-lapse video (bright-field and fluorescence imaging) shows 4 days of epithelial morphogenesis in fibroblast (cyan)-organoid (red) co-culture (day 0–4). Scale bar: 100 μm. Snapshots from the movie are depicted in Fig 3A.
(AVI)

**S3 Movie. Fibroblasts form close contacts with epithelium in the organoid branching points.** 3D structure of organoid-fibroblast interaction. Single images are shown in Fig 3D. Luminal cells (KRT8), red; basal cells (KRT5), blue; all cells (F-actin), white.
(AVI)

**S4 Movie. *Myh9* knock-down in fibroblasts decreases their morphogenetic potential.** Time-lapse videos show 5 days of epithelial morphogenesis in co-culture with either control (left) or *Myh9* knocked-down fibroblasts (siRNA-mediated knockdown; si*Myh9*; right). Time is in hours. Snapshots from the video are depicted in Fig 4.
(AVI)

**S5 Movie. *Myh9* knock-out in fibroblast decreases their morphogenetic potential.** Time-lapse video captures 4 days of epithelial morphogenesis in fibroblast (cyan)-organoid(red) co-culture with either control (Ad-GFP; left) or *Myh9* knocked-out fibroblasts (adeno-Cre-mediated knock-out; Ad-Cre-GFP; right). Snapshots from the video are depicted in Fig 4. Scale bar: 100 μm.
(AVI)

**S6 Movie. Fibroblasts organization around bifurcating TEB.** Z-stack scroll-through of mammary gland whole-organ imaging, showing a bifurcating TEB. DAPI in blue, vimentin in white, tdTomato in red. MIP and appropriate scale bar are depicted in Fig 6.
(AVI)

**S7 Movie. Fibroblasts organization around invading TEB.** Z-stack scroll-through of mammary gland whole-organ imaging, showing an invading TEB. MIP and appropriate scale bar are depicted in Fig 6.
(AVI)

**S1 Data. Excel spreadsheet with individual numerical data underlying plots and statistical analyses.** The data are organized into separate sheets corresponding to the following figure panels: 1B, 1C, 1D, 1E, 2B, 2D, 3C, 3G, 4D, 4F, 4H, 4I, 5B, 5C, 5F, 5G, 6B, S2C, S2E, S3B, S4B,

S4D, S5C, S5D, S6B, S6C, S6D, S7A, S8B, S8C, S9B, S10C, S10D, S10G, and S11B.
(XLSX)

**S1 Table. The list of pharmacological and viral compounds.**
(DOCX)

**S2 Table. The list of detection agents used in this study.**
(DOCX)

**S3 Table. The list of primers used for qPCR in this study.**
(DOCX)

## Acknowledgments

We are grateful to Danijela Matic Vignjevic for critical review of the manuscript, to Denisa Belisova for mouse husbandry, and to Maria Luisa Martin Faraldo for the LAMA5 antibody. We are thankful to Enantis for providing FGF2 and FGF2-STAB. We acknowledge the core facility CELLIM of CEITEC, supported by the Czech-BioImaging large RI project (LM2023050 funded by MEYS CR), for their support with obtaining scientific data presented in this paper. We gratefully acknowledge the Cell and Tissue Imaging Platform (PICT-IBiSA) at Institut Curie, member of the French National Research Infrastructure France-BioImaging (ANR-10-INBS-04).

## Author Contributions

**Conceptualization:** Jakub Sumbal, Zuzana Sumbalova Koledova.

**Funding acquisition:** Jakub Sumbal, Silvia Fre, Zuzana Sumbalova Koledova.

**Investigation:** Jakub Sumbal, Zuzana Sumbalova Koledova.

**Methodology:** Jakub Sumbal, Zuzana Sumbalova Koledova.

**Project administration:** Zuzana Sumbalova Koledova.

**Resources:** Silvia Fre, Zuzana Sumbalova Koledova.

**Supervision:** Silvia Fre, Zuzana Sumbalova Koledova.

**Validation:** Jakub Sumbal.

**Writing – original draft:** Jakub Sumbal, Zuzana Sumbalova Koledova.

**Writing – review & editing:** Jakub Sumbal, Silvia Fre, Zuzana Sumbalova Koledova.

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
