## [Editor Report · Decision Letter 0]

22 Mar 2023

Dear Dr Koledova, 

Thank you for submitting your manuscript entitled "Fibroblast-induced mammary epithelial branching depends on fibroblast contractility" for consideration as a Research Article by PLOS Biology.

Your manuscript has now been evaluated by the PLOS Biology editorial staff as well as by an academic editor with relevant expertise and I am writing to let you know that we would like to send your submission out for external peer review.

Once your full submission is complete, your paper will undergo a series of checks in preparation for peer review. After your manuscript has passed the checks it will be sent out for review. To provide the metadata for your submission, please Login to Editorial Manager (https://www.editorialmanager.com/pbiology) within two working days, i.e. by Mar 24 2023 11:59PM.

Kind regards,

Ines

--

Ines Alvarez-Garcia, PhD

Senior Editor

PLOS Biology

---

## [Decision Letter · Decision Letter 1]

17 Jul 2023

Dear Dr Sumbalova Koledova,

We have now had the chance to discuss your revision plan of your manuscript entitled "Fibroblast-induced mammary epithelial branching depends on fibroblast contractility" which was peer-reviewed at PLOS Biology.

We agree that the experiments you mention seem to go a long way to address the concerns previously raised by the reviewers, thus we would like to invite you to submit a revision that thoroughly addresses the reviewers' reports.

Given the revision needed, we cannot make a decision about publication until we have seen the revised manuscript and your response to the reviewers' comments. Your revised manuscript is likely to be sent for further evaluation by all or a subset of the reviewers.

We expect to receive your revised manuscript within 3 months, but please email us (plosbiology@plos.org) if you have any questions or concerns, or would like to request an extension. At this stage, your manuscript remains formally under active consideration at our journal; please notify us by email if you do not intend to submit a revision so that we may withdraw it.

**IMPORTANT - SUBMITTING YOUR REVISION**

3. Resubmission Checklist

a) *PLOS Data Policy*

b) *Published Peer Review*

Sincerely,

Ines

--

Ines Alvarez-Garcia, PhD

Senior Editor

PLOS Biology

Reviewers' comments

Rev. 1:

In this highly interesting and intriguing manuscript, Sumbal et al. describe how mammary fibroblast change the behavior of mammary epithelial organoids and induce budding/branching in the absence of commonly used "branching inducers" such as Fgf2. This induction was dependent on the physical contact between fibroblasts and mammary epithelial cells, as well as fibroblast contractility. Thereafter, the authors provide evidence that this fibroblast-dependent branching behavior is associated with patterned YAP and ERK activity.

This study is thought provoking and proposes an interesting novel concept. The finding that co-culture of contractile fibroblasts in breast cancer spheroids (that normally do not branch) reconstitutes budding morphogenesis nicely substantiates and expands the findings made with primary mammary organoids. Disappointingly, however, it remained unclear how well the proposed mechanism reflects branching morphogenesis in vivo. Also, certain aspects of the study lack quantitative analysis that are needed to support the conclusions.

Major issues:

My main concern is to what extent the observed phenomenon recapitulates in vivo situation, as this remains completely open. Some in vivo analysis would greatly strengthen the manuscript to support the main conclusion and to reduce the likelihood of an in vitro artefact. For example, how are fibroblasts located and organized in vivo in growing and bifurcating TEBs? Or in side branches? Can 'fibroblast loops' be observed in vivo?

The authors make the point that the organoids cultured with fibroblasts retain a bilayer structure (though without full basal cell coverage, if I understood it correctly (Figure 3C), hence not recapitulating the in vivo epithelium in this respect). However, TEBs are multicellular stratified structures, yet the authors discuss that their model is a model for TEB bifurcation. To me this is counter-intuitive given the very different epithelial structure of TEBs and the organoids described in this study. What evidence can the authors provide to support their conclusion that their system models TEB clefting?

Specific comments:

1. Lines 138-139 state: "On the branched organoids, fibroblasts were exclusively located around the necks of the nascent branches … (Figure 3B)."

Lines 255-256: "Our work reveals that mechanical strain imposed on mammary epithelial cells by fibroblasts results in epithelial folding with negative curvature in the epithelial-fibroblast contact points."

These are strong statements, however, these conclusions are not evident to me based on the images and should be backed-up with quantifications on fibroblast locations with respect to the co-cultured organoids. In fact, Movie 1 rather seems to suggest that fibroblasts arrive once the negative epithelial curvature is already emerging. Can the authors exclude the possibility that fibroblasts stabilize, rather than induce the formation of new buds? Quantification of the fibroblast locations in fixed samples and/or quantifications of their behaviors in the time-lapse movies would be needed to support these conclusions that are critical to the manuscript.

2. Functional data on the importance of fibroblast contractility (Fig. 4C-E) is convincing. Yet, contractility could also be related to cell motility. I see many of the fibroblasts "swimming" toward the epithelium in the videos provided, but what about the Myh9 ablated ones? Can the authors exclude the possibility that they fail to move toward the organoid? A movie similar to Movie 2 but using Myh9 deleted fibroblasts would be informative (fibroblasts visualized e.g. using live dyes). An alternative option would be to perform live imaging of organoids that are already undergoing branching followed by blebbistatin/ROCK inhibitor treatment and further imaging. How does the perturbation change fibroblast behaviors?

The authors also describe "cellular loops" generated by fibroblasts (Figure 4A). However, Fig. 4A looks somewhat different from Fig. 3B that also shows the location of fibroblasts. Could the authors provide quantifications on how often such 'cellular loops' were detected?

3. The authors make the point that "fibroblast-induced branching requires epithelial proliferation". I am not fully convinced by the importance of these findings (Fig. 5E-F) - doesn't Fgf2 -dependent branching also require proliferation? Aren't new cells needed to build new branches?

The authors also make the point of patterned cell proliferation and show that the stalk is more proliferative compared to the tip. Do the authors consider this a physiologically relevant finding? In other words, does this fit with the data from TEBs?

If patterned proliferation is linked with presence of fibroblasts, then presumably it is not observed in Fgf2-induced branches? This should be assessed.

4. The authors also show that YAP and ERK activity are patterned and based on the absence of patterning in co-cultures with Myh9-deleted fibroblasts, it is concluded that this depends on fibroblast contractility. However, under these conditions, branching does not take place, so this experimental approach is perhaps not the most informative one. What about Fgf2-induced branching: are YAP and ERK activities patterned differently compared to fibroblast co-cultured organoids?

5. The authors propose that the underlying epithelial cells can sense the contact with contractile fibroblasts as a mechanical stress leading to specific nuclear accumulation of YAP in the neck region of the buds. I am not sure if I am misinterpreting Figure 5I (a higher resolution close-up would be informative), but to me it looks there is a lot of nuclear YAP in organoids cultured without fibroblasts and Fgf2, which would argue that there is no such relationship between contractile fibroblasts and epithelial YAP. Instead, nuclear YAP might simply reveal and reflect the proliferative status of epithelial cells.

Minor comments:

1. Line 86: "Organoids co-cultured with fibroblasts developed bigger but less numerous branches (Figure 1A, C)." It remained unclear to me where the size of the organoids was reported.

2. The results and conditions used n Fig. 2 are very clear. However, thereafter it remained unclear whether the fibroblasts used in the experiments were preaggregated or not.

3. The correlation coefficient (linear regression) should be reported.

4. Movie 2 could be more informative if it showed one branching event from the beginning to the end.

Rev. 2:

This paper describes the morphogenesis of mammary organoids with respect to branching morphogenesis evoked by co-culture of fibroblast or incubation with the growth factor FGF2. They propose that fibroblasts shape branching morphogenesis by interacting with the organoids and providing an actomyosin-based constriction mechanism that participates in the shaping of the branches.

My first criticism is that the paper is not well written. As a reader with limited experience in mammary morphogenesis, I had the feeling that the authors expected me to know his organoid system as well as they do it. By example, the existence of myoepithelila cells in these organoid cultures (absent from simple epithelial cyst models such as MCF10A or MDCK) was never mentioned before discussed in the results (and this is clearly a strength of this model). The data in the paper are of good quality, but the different experiments are anecdotal and do not culminate in a paper with a clear message. Many perturbation experiments (such as the proliferation experiments in figure 5) cannot lead to the conclusions that the authors make. The experimental design is wrong from the beginning, leading to data that can only be mis- or over-interpreted. Finally, the discussion is hand waving, and the many results are not even discussed (the presumable interaction between the fibroblast and the myoepithelial cells). The finding that fibroblast contractility is required for fibroblast constriction around epithelia is trivial. I think it is the striking interaction between the fibroblasts and the epithelium that is interesting, and should have been studied.

To me the paper is not of interest to the wide audience of PLOS biology. I would suggest to the authors to change the narrative of the paper to display the results (the images are of high quality) and send this to a specialized journal, or maybe PLOS one with major revisions.

Below some more specific comments.

Figure 1F is mentioned after 1G, cite figure panels in order of appearance

Figure 1G: denote what asterisks point to !

Line 127: Ras is a GTPase and not a kinase ! Usually one refers to the GTPase by mentioning its oncogenic mutation, as well as the Ras isoform: by example KRas G12V ?

Figure 3: the panels refer to mtomato and mGFP, the legend to tdtomato and gfp. What does mtomato or mGFP refer to ? Monomeric tomato ? Do the authors use a monomeric or a tandem tomato (I think monomeric tomtato does not even exist ?).

Figure 3B: can the authors tell us what they see in panels 1,2,3. Are they just examples, or do the authors want to show us fibroblast - organoid interactions with specific features ! This reviewer feels like he has to guess what the authors want to tell him.

"On the branched organoids, fibroblasts were exclusively located around the necks of the nascent branches and sat directly in contact with the epithelium (Figure 3B). Is there a statistic about this ?

Figure 3C: interaction of fibroblasts with krt5 positive myoepithelial cells. It is difficult to judge if this interaction is just anecdotal, or a real phenomenon with n=1 example. Can the authors show us a statistic. The images are complex to understand.

Also, not being familiar with the organoid culture system, I was very surprised at this point in the paper to learn about the presence of myoepithelial cells and not only epithelial cells in this system. I think it would be worthy to mention this in the introduction (this is a clear strength of this model system !). If myoepithelial cells are important for organoid - fibroblast interactions, then I imagine that fibroblasts must interact differently with MCF7 -ras cells that I believe do not have such myoepithelial cells ? (or maybe I missed something ?).

Figure 3D: I am totally confused here ! Can the authors share with me how they identify epithelial cells from myoepithelial cells in EM pictures ? Maybe it is totally obvious to them, but as a naïve reader, I was not even told in the introduction of the paper that this system had myoepithelial cells ! This should be written to be accessible to everybody.

What are the white arrows referring to ? I could not find this info in the legend ?

Figure 3F: is this type of quantification really useful ? I think one can see that the cells bind the laminin matrix on the epithelial cells.

Line 152 "Moreover, the fibroblasts expressed phosphorylated myosin light chain 2 (P-MLC2), a marker of active non-muscle myosin II (Figure 4B)." A cell does not express p-MLC2, it phosphorylates MLC2!

Figure 4B: yes, there is as expected pMLC2 in epithelia as well as in fibroblasts! Both cell types display contractile structures !

Figure 4C: it is difficult to interprete this experiment since inhibition of ROCK or MLC will indeed affect both epithelial fibroblast cells.

"178 To test whether epithelial proliferation (and thus expansion) plays a role in organoid branching in

179 co-cultures, we inhibited cell proliferation using aphidicolin (DNA polymerase inhibitor), upon which

180 we observed a severe defect in organoid branching (Figure 5D-F). To test for the possibility that the

181 observed effect could be caused by inhibition of fibroblast proliferation, we performed the experiment

182 also with fibroblasts pretreated with mitomycin C, an irreversible DNA synthesis blocker (Figure 5D).

183 The pretreatment of fibroblast with mitomycin C had no effect on the result (Figure 5D-F),

184 demonstrating that fibroblast proliferation is dispensable while epithelial proliferation is necessary for

185 organoid branching in co-cultures."

I think the authors should really think about their experiments. The perturbations they use will not only induce loss of cell proliferation but also cell death in both cell populations. The epithelium will react with an epithelial homeostasis response. I think the kind of causalities the authors are trying to make are just impossible !

Figure 5G: my eyes cannot see the gradient of ERK that the authors see. In panel 5 from the left, I see kind of a gradient of ERK on the left branch of the epithelial bud, but not on the right branch ! The nuclear stain indicates that the right branch is on the right focal plane. It is difficult to conceive how the authors can get a robust statistic with such images. Or then there is a more complex pattern not fully understood ?

Also, work from the matsuda lab suggests the existence of discrete ERK pulses, that one could from the image in fig. 5I - P-ERK, in which you can clearly see some ON and OFF cells. So the idea of a gradient of p-ERK signaling is not intuitive.

---

## [Decision Letter · Decision Letter 2]

2 Nov 2023

Dr Sumbalova Koledova,

Thank you for your patience while we considered your revised manuscript "Fibroblast-induced mammary epithelial branching depends on fibroblast contractility" for publication as a Research Article at PLOS Biology. This revised version of your manuscript has been evaluated by the PLOS Biology editors, the Academic Editor and the two original reviewers.

Based on the reviews (attached below), we are likely to accept this manuscript for publication, provided you satisfactorily address the remaining points raised by Reviewer 1. Please note, however that after discussing the points with the Academic Editor, we would like to you to keep the Yap data in the manuscript. Please also make sure to address the following data and other policy-related requests stated below.

We expect to receive your revised manuscript within two weeks. 

*Published Peer Review History*

*Press*

Sincerely,

Ines

--

Ines Alvarez-Garcia, PhD

Senior Editor

PLOS Biology

DATA POLICY:

Many thanks for sending us the data underlying all the graphs shown in the figures. The data files should be saved using the following convention: S1_Data.xlsx (using an underscore).

In addition, please mention in each corresponding figure legend where the data can be found and to name them in the text, please use S1 Data. For example, you can add at the end of each figure legend that contains graphs the following sentence: "The data underlying the graphs shown in the figure can be found in S1 Data."

SPECIES INDICATED IN THE ABSTRACT? 

- Please note that per journal policy, the model system/species studied should be clearly stated in the abstract of your manuscript. Please mention that the in vivo experiments have been performed in mice.

Reviewers' comments

Rev. 1:

The authors of this manuscript have made substantial efforts to improve the manuscript and provide a wealth of additional data that substantially strengthens the paper.

Here I will comment the responses provided to my main criticism:

1. In vivo importance of the observed phenomenon.

To address this issue, the authors now provide 3D images of pubertal glands showing that the necs of growing TEBs are associated with contractile (aSMA+) fibroblasts and that bifurcating tips are associated with "looping" fibroblasts. While the images provided are gorgeous, and undoubtedly motivate the authors to continue their research toward in vivo analyses and approaches, without quantifications these new images remain anecdotal (at which point of TEB bifurcation they appear, are they associated with all bifurcating TEBs, is there truly a loop - the video of confocal stacks suggests that the loop is only partial, not extending the entire circumference of the nascent new tip etc.).

It is fully understandable that within the revision timeframe obtaining enough material for quantification may not be feasible, but this also means that the authors have to remain cautious about their conclusions. This works mostly throughout the manuscript, yet e.g. the very last concluding chapter of the Discussion includes an overinterpretation: "In conclusion, we find that fibroblasts drive branching morphogenesis of the mammary gland by exerting mechanical forces on epithelial cells." This is a fair conclusion on the organoid data, but the new in vivo images are not sufficient to justify this conclusion on the pubertal mammary gland, but obvious it is a good basis for a hypothesis that this might be the case.

I also think that the authors should discuss their fibroblast looping model in light of the recent paper from the Ewald lab (PMID: 36602106) showing that genuine tip bifurcations can occur also in the classic organoid set-up, i.e. without fibroblasts (while to me it still remains unclear to which extent in the organoids - with or without fibroblasts - branching really recapitulates true bifurcation. Or are majority of the branching events in fact 'bulging/blebbing' of the bilayered stalk once it has proliferated sufficiently to have enough cells for a new branch?). Similarly, another group of has shown that in embryonic mammary organoids, bifurcations may occur in the absence of stromal cells (PMID: 37367826).

2. Is the organoid model a good model of TEBs due to the different cellular architecture (TEBs being multilayered, organoids bilayered)?

The authors now provide new data with FGF2-STAB + fibroblasts and show that this leads to a more multicellular structure and still the fibroblasts appear to loop the same way and lead to faster onset of budding/branching that FGF2-STAB alone (which also induces budding/branching, but apparently only after 4 days of culture). This is a nice addition to the paper but does not exclude the fact that majority of the fibroblast data is from bilayered organoids. Again, I call after caution in data interpretation.

3. Somewhat scarce quantification of a number of different data in the original version of the manuscript.

The authors are to be applauded for their work on fixing this issue - the manuscript is much stronger now than it was.

4. The question on contractility vs. motility

This question has been addressed appropriately. The finding that fibroblasts may also play a role branch stabilization is a nice addition to the manuscript.

5. Importance of patterned cell proliferation in organoids.

The authors give a plausible explanation for the difference between organoids and TEBs, but again, this is one additional difference between them, further warranting caution in overinterpreting the in vivo relevance of the organoid data.

6. YAP

I remain confused about YAP and its potential link with the contractile fibroblasts.

The authors write" …indicating that YAP signal arises from the contact with contractile fibroblasts and not the overall shape of the epithelial branch" and "Knockout of Myh9 in fibroblasts prevented YAP pattern formation (Figure 4J), indicating that contact with contractile fibroblasts regulates YAP epithelial distribution."

Am I interpreting Fig. 4J correctly? To me it looks that in organoids with Myh9 deleted fibroblasts, YAP is high in all cells. So how can "YAP signal arise from the contact with contractile fibroblasts"?

I think this is the weakest part of the manuscript, and I recommend the authors to consider removing it from the manuscript. Previous (PMID: 24589775) and recent (https://www.biorxiv.org/content/10.1101/2023.08.23.554465v1.full) reports indicate that YAP is highest in growing TEBs, which according to the current manuscript are largely devoid of fibroblasts (except upon bifurcation).

So, to me the correlation between YAP and proliferation appears much stronger than between YAP and contractile fibroblasts.

7. One final point about Discussion. There, the authors speculate that "in vivo the highly dynamic mechanically active fibroblasts could initiate formation of epithelial clefts and further reinforce them by subsequent deposition and remodeling of ECM." I wonder if the authors would like to ponder this speculation in light of the evidence indicating that pubertal branching morphogenesis is highly stochastic, following "a branching and annihilating random walk" model (PMID: 28135720).

Rev. 2:

The authors have performed a lot of experiments to satisfy my concerns, as well as those of the other reviewer. Most importantly, they have clarified a lot of points that make the manuscript more accessible. I think it was a good decision to leave out the MAPK dataset that is difficult to interpret. I think that the paper is a good example of next generation organoid in which tackles how multiple cell types contribute to organogenesis. I am happy to accept the manuscript, and congratulate the authors.

---

## [Editor Report · Decision Letter 3]

24 Nov 2023

Dear Dr Koledova,

Thank you for the submission of your revised Research Article entitled "Fibroblast-induced mammary epithelial branching depends on fibroblast contractility" for publication in PLOS Biology. On behalf of my colleagues and the Academic Editor, Emma Rawlins, I am delighted to let you know that we can in principle accept your manuscript for publication, provided you address any remaining formatting and reporting issues. These will be detailed in an email you should receive within 2-3 business days from our colleagues in the journal operations team; no action is required from you until then. Please note that we will not be able to formally accept your manuscript and schedule it for publication until you have completed any requested changes.

PRESS

Sincerely, 

Ines

--

Ines Alvarez-Garcia, PhD

Senior Editor

PLOS Biology
